# Asbestos and Iron

**DOI:** 10.3390/ijms241512390

**Published:** 2023-08-03

**Authors:** Andrew J. Ghio, Matthew Stewart, Rahul G. Sangani, Elizabeth N. Pavlisko, Victor L. Roggli

**Affiliations:** 1US Environmental Protection Agency, Research Triangle Park, NC 27711, USA; 2Environmental Health and Engineering, Johns Hopkins Bloomberg School of Public Health, Baltimore, MD 21205, USA; mstewa12@jhu.edu; 3Department of Medicine, West Virginia University, Morgantown, WV 26506, USA; rgsangani@hsc.wvu.edu; 4Department of Pathology, Duke University Medical Center, Durham, NC 27710, USA; elizabeth.pavlisko@duke.edu (E.N.P.); victor.roggli@duke.edu (V.L.R.)

**Keywords:** asbestos, iron, lung diseases, alveolar macrophages, ferritin

## Abstract

Theories of disease pathogenesis following asbestos exposure have focused on the participation of iron. After exposure, an open network of negatively charged functional groups on the fiber surface complexes host metals with a preference for iron. Competition for iron between the host and the asbestos results in a functional metal deficiency. The homeostasis of iron in the host is modified by the cell response, including increased import to correct the loss of the metal to the fiber surface. The biological effects of asbestos develop in response to and are associated with the disruption of iron homeostasis. Cell iron deficiency in the host following fiber exposure activates kinases and transcription factors, which are associated with the release of mediators coordinating both inflammatory and fibrotic responses. Relative to serpentine chrysotile, the clearance of amphiboles is incomplete, resulting in translocation to the mesothelial surface of the pleura. Since the biological effect of asbestos is dependent on retention of the fiber, the sequestration of iron by the surface, and functional iron deficiency in the cell, the greater clearance (i.e., decreased persistence) of chrysotile results in its diminished impact. An inability to clear asbestos from the lower respiratory tract initiates a host process of iron biomineralization (i.e., asbestos body formation). Host cells attempt to mobilize the metal sequestered by the fiber surface by producing superoxide at the phagosome membrane. The subsequent ferrous cation is oxidized and undergoes hydrolysis, creating poorly crystalline iron oxyhydroxide (i.e., ferrihydrite) included in the coat of the asbestos body.

## 1. Introduction

Asbestos is an industrial term used to denote a group of six fibrous, silicate particles previously of commercial value because of their tensile strength and chemical and heat resistance. These fibrous particles include five amphibole minerals (actinolite, amosite, anthophyllite, crocidolite, and tremolite) and one serpentine mineral (chrysotile) (Table 1).

Asbestos has been used by humans for thousands of years. The commercial mining of large deposits of chrysotile began in the 1870s and extensive exploitation of amosite and crocidolite followed soon after. However, several decades after initiating its utilization in a myriad of applications, reports of asbestos-related human disease (asbestosis, lung and laryngeal cancers, and mesothelioma) appeared. A description of pulmonary fibrosis after occupational exposure to asbestos was originally provided to the British parliamentary committee in 1906; this was later designated asbestosis [1]. In 1955, epidemiologic studies initially established an association between lung cancer and asbestos exposure [2,3]. Cases of mesothelioma (pleural) were first observed following asbestos exposure in South Africa in 1960 [4]. Almost all use of asbestos was subsequently stopped in most nations of the world (e.g., after 1989 in the United States).

The pathogenesis of disease following exposure to asbestos has not been fully determined. A differential in their biological effects revealed that amphiboles have a greater capacity to cause asbestosis, lung cancer, and mesothelioma relative to chrysotile. It was recognized that there were greater quantities of iron included in some commercial amphiboles, specifically amosite and crocidolite. Accordingly, theories of disease pathogenesis following asbestos exposure were developed which focused on the participation of iron. There is support for metals’ involvement in the biological effect of asbestos, with iron chelation therapy and phlebotomy demonstrating possible preventive effects against asbestos-induced carcinogenesis in animal models [5,6]. Postulates regarding the pathogenesis of asbestos-related disease include oxidative stress mediated by the in vivo production of hydroxyl radicals by iron and, more recently, accompanied by ferroptosis [7,8]. An alternative hypothesis was submitted stating that asbestos-related disease reflected the disruption of host iron homeostasis following fiber exposure [9]. The latter suggests that the pathogenesis of asbestos-related disease is dependent on the complexation chemistry of iron rather than the oxidant chemistry. Accordingly, the relationship between asbestos and iron was examined.

## 2. The Crystal Lattice of Asbestos and Structural Iron

Amphibole fibers are double-chain, tetrahedral silicates with a rod-like structure. The single serpentine fiber, chrysotile, is composed of sheets of silicon-centered tetrahedra joined to sheets of octahedral coordinated magnesium (brucite or Mg(OH)_2_) in a flexible structure [10]. Chrysotile fibrils are curved either concentrically or spirally into tubular structures [11]. The determination of iron in standard reference samples of asbestos revealed that their concentrations in amosite and crocidolite (28% and 27%, respectively) were greater than those in chrysotile (0–2%) [12]. Metal constituents of asbestos are not always structurally incorporated in the fiber but can be present as contaminating minerals such as magnetite (Fe_3_O_4_), brucite, or chromite (FeCr_2_O_4_). In asbestos with iron in the ideal molecular formula (i.e., actinolite, amosite, and crocidolite), structural metal is found in octahedral sites. In asbestos without iron in the ideal molecular formula (i.e., anthophyllite, tremolite, and chrysotile), the metal is also frequently observed in small quantities at octahedral sites after isomorphic replacement (e.g., iron cations at magnesium and/or silicon sites of the chrysotile sheet). This is comparable to aluminum, which can also be a common foreign element in fibers [13].

It has been proposed that the biological activity of amphiboles could be greater than that of chrysotile after the mobilization of a greater concentration of its structural iron which possibly supports electron transport and an oxidative stress (e.g., hydroxyl radical). Structural iron at the exposed surface of a mineral can be mobilized, or extracted, under specific conditions, but this will contribute to fiber destruction (e.g., weathering) [14,15]. The involvement of structural iron in oxidative stress and biological effects by asbestos would subsequently be dependent on fiber destruction [16]. Asbestos in the lower respiratory tract is almost exclusively observed within macrophages [17]. The phagocytosis of fibers by macrophages creates a vacuole (a phagosome/phagolysosome), with the internalized asbestos being exposed to lower pH values (e.g., 4–4.5). As a result of the lack of acid-soluble surface groups, the amphiboles have negligible solubility at any pH that might be encountered in a living system [18] (Figure 1A). In contrast, the phagocytosis of chrysotile is associated with dissolution of the fiber [19] (Figure 1B). The brucite layer (i.e., magnesium hydroxide) of chrysotile is dissolved in an acid milieu such as that in a macrophage [20,21,22,23].

However, after inducing radioactivity in chrysotile using neutron irradiation, there was very little translocation of any metal from the asbestos in an animal model, indicating that metal mobilization with in vivo dissolution was either an extremely slow process or did not occur [24]. Dissolution of chrysotile great enough to impact available iron is not predicted, as there are numerous compounds in human cells and tissues with the capacity to complex/chelate small concentrations released from the asbestos (e.g., apotransferrin, apolactoferrin, carboxylates, and phosphates) [16]. In addition, based on the ideal molecular formula, not all chrysotile will include iron. Finally, the dissolution of structural iron in the crystal lattice of asbestos would suggest a greater biological effect of chrysotile relative to the amphiboles, which is contradictory to what is observed.

The current evidence does not support that the biological effect of asbestos is impacted by structural iron in a crystal lattice. The involvement of structural iron in oxidative stress could not explain the biologic activity of (1) amphiboles that do not have any iron included in their ideal molecular formula (e.g., tremolite) and (2) other particles/fibers that may or may not include structural iron. In addition, among those particles/fibers with the greatest associations with human disease, there are numerous examples with no structural iron (i.e., erionite and quartz, which have ideal molecular formulae of (Na_2_,K_2_,Ca,Mg)_4.5_Al_9_Si_27_O_72_ and SiO_2_, respectively) while benign particles can demonstrate a significant content of iron (e.g., illite, (K,H_3_O)(Al,Mg,Fe)_2_(Si,Al)_4_O_10_[(OH)_2_,(H_2_O)]).

## 3. The Surface of Asbestos and Complexed Iron

Interactions between a mineral and its environment occur at the surface. Such interactions include adsorption and cation exchange (as well as dissolution, precipitation, and oxidation/reduction). The cleavage of an oxide mineral exposes individual atoms of (1) oxygen and (2) metal at the surface wherever the fracture cuts across a chemical bond. Water reacts with this surface by adding (1) hydrogen to the oxygen atoms and (2) hydroxyls to the metals. Subsequently, following the dissociative chemisorption of water, the surface of a metal oxide is hydroxylated. Each of these hydroxides can donate a proton, and these negatively charged, surface functional groups then bind positively charged atoms or molecules. There will be some exchange between the asbestos and cations in any surrounding environment. The resultant exchange capacity is a measure of how many cations can be retained on the surface. In oxides, this cation-exchange capacity can directly correlate with the surface hydroxyl site density [25].

All asbestos surfaces have some concentration of silanol groups (-Si-OH), which have a significant ionic character. Silanol dissociation contributes to a net negative charge on the surface that generates a capacity for adsorption and the exchange of cations. The open network of negatively charged silanol groups on the fiber surface presents spaces large enough to accommodate adsorbed metal cations. The distribution and concentration of the silanol groups will reflect the specific, and possibly unique, fracture cut. As a result of its electropositivity, Fe^3+^ has a high affinity for oxygen-donor ligands [26]. Consequently, this metal will react with silicate surfaces. Dose-dependent complexation of inorganic iron has been demonstrated for crystalline silicates with critical stability constants (Ksc) up to 1 × 10^17^ [27,28]. While a hydroxylated surface complexes numerous different cations, a preference for iron is frequently demonstrated [29,30,31,32].

## 4. Asbestos, Cell Iron Homeostasis, and Their Biological Effects

Iron is essential for almost every organism, and its lack of availability can restrict life in environments ranging in size from the ecosystem of the Pacific Ocean to a bacterium [33,34]. Its unique coordination chemistry led to its evolutionary selection for numerous essential cell functions [35]. Iron concentrations that are inadequate to meet the requirements for life necessitate the development of pathways to acquire this critical metal. Concurrently, the metal-catalyzed generation of radicals can potentially present oxidative stress. Consequently, cell iron homeostasis, including the import, storage, and export of the metal, is carefully regulated, and life exists at an interface between iron deficiency and sufficiency. If a cell reaches a lower threshold of iron concentration, there is obstruction of the cell cycle and an initiation of regulated cell death [36,37,38].

The surface of asbestos can acquire iron. The binding of extracellular iron sources by any particle/fiber has not been demonstrated and is unlikely to occur as a result of the extremely strong binding of the metal by host molecules (e.g., the Ksc of transferrin for iron is approximately 10^20^ M^−1^ at pH 7.4). Intracellular iron levels available for complexation by the fiber surface can approach the concentration of the labile iron pool (less than 1–5 µM) [39,40]. Accordingly, functional groups at the fiber surface will compete for iron utilized by the host cell in functions critical for survival (Figure 2).

Mitochondrial iron concentrations decrease after the exposure of respiratory cells to asbestos as the surface sequesters this host metal [41]. Other intracellular sources of iron are also available to asbestos, but it is difficult to quantify their utilization. With the binding of the host metal by the fiber surface, the cells exposed to asbestos must perceive a functional iron deficiency (Figure 3).

Specific cell functions can be compromised by asbestos exposure unless concentrations of iron are re-established. Iron homeostasis in the host must be modified and the cell response will include the upregulation of import to correct the functional deficiency following loss of the metal to the fiber surface [42,43]. The expression of divalent metal transporter 1 (DMT1) is increased after asbestos exposure, reflecting the diminished iron concentrations available in the cells [44,45]. Increased metal import will successively follow asbestos exposure and is required for cell survival [45]. The increased concentration of imported metal causes an elevated expression of ferritin, which is also observed to be elevated following asbestos exposure [41,45]. As a result of the interaction between the cell and the fiber, adjusted iron homeostasis occurs and results in some quantity of metal being complexed at the surface of the endocytosed fiber, increased total cell concentrations, elevated cell ferritin levels, and sufficient cell and mitochondrial iron to meet the requirements for continued function. While the fiber continues to bind some portion of the iron, metal concentrations are modified after exposure to make continued cell function and survival feasible.

The host also attempts to re-acquire its metal that has been directly sequestered by the fiber. The movement of iron necessitates ferrireduction to chemically reduce the metal to a ferrous state (e.g., transmembrane transport). This chemical reduction is frequently accomplished by superoxide generated in the cell (e.g., membranes of the cell, mitochondria, and lysosomes). The increased availability of iron (i.e., cell pretreatment with FAC) will subsequently diminish any functional iron deficiency and superoxide production following asbestos exposure [41]. The “oxidative stress” after asbestos exposure includes mitochondria as a source since rotenone inhibits some portion of superoxide generation and iron uptake by the cell. The electron transport chain in the mitochondria can be a major source of superoxide production [46,47]. Through a comparable pathway, cellular superoxide generation can increase after exposure to iron chelators [48,49]. The generated superoxide functions in ferrireduction in cells’ acquisition, transport, and translocation of iron. Such ferrireduction is an essential, and frequently limiting, reaction in iron acquisition, transport, and translocation [50,51].

The biological effects of asbestos develop in response to and are associated with the disruption of iron homeostasis (Figure 3 and Figure 4).

Inflammation is included in the host response to functional iron deficiency. The complexation of host iron by asbestos initiates pathways, which culminate in the release of inflammatory mediators and inflammation. Cell iron deficiency following exposure to fibers activates kinases and transcription factors, which are associated with the release of inflammatory mediators. Cell exposure to asbestos activates mitogen-activated protein (MAP) kinases, and a portion of this response can be diminished by increasing the concentration of available iron [52,53,54]. Transcription factors involved in the expression of inflammatory mediators are also activated by particle/fiber exposure [55,56,57,58]. Many of these same transcription factors control cell death, and if the response to correct the functional iron deficiency is insufficient, some form of apoptosis will ensue [59,60,61,62]. Comparable to MAP kinases, increased iron availability decreases the activation of transcription factors following exposure to particles/fibers [53,54,63]. After impacting cell signaling and transcription factors, asbestos exposure will produce changes in the expression of inflammatory mediators. Changes in protein expression for IL-6 and IL-8 after asbestos exposure are diminished by cell pretreatment with iron, reflecting their association with metal availability [41]. It is the functional iron deficiency after asbestos exposure that activates kinase signaling and transcription, increases the release of mediators, and coordinates the inflammatory response.

Pulmonary fibrosis is also included in the host response to reverse the loss of essential metal after its sequestration by asbestos (Figure 3 and Figure 4) [64]. Fibrosis is characterized by the excessive accumulation of extracellular matrix (ECM). Collagen, frequently a main component of ECM, includes carboxylates, hydroxyls, and amines, which complex iron at regular intervals, increasing its availability; collagen peptides (<10 kDa) demonstrate comparable metal binding activity [65,66,67,68,69,70]. This interaction of collagen with metals is a recognized method for its stabilization (i.e., tanning, which most commonly is achieved with chromium, but iron can be employed) [71,72]. Elastic tissue, a second component of ECM, also stains with iron compounds, demonstrating an affinity for complexing metal [73,74,75]. Polyuronates are a major component of glycosaminoglycans (GAGs) in ECM. Hyaluronic acid is the most abundant GAG in ECM and forms a coordination complex with metals including iron [76,77,78]. Higher doses of iron decrease the synthesis of these biopolymers (collagen, elastin, and polyuronates) supporting their role in metal homeostasis [79,80,81,82]. Metals also participate in the depolymerization of these same ECM components and, following the reaction with iron, the polymer (i.e., collagen, elastin, and GAG) will be degraded [76,83,84]. The depolymerization of the biopolymer improves the cell import of metal. Accordingly, functional iron deficiency after asbestos is proposed to increase the synthesis of the biopolymers included in fibrosis. These materials complex metal, and their depolymerization then provides oligomers with bound iron to cells via receptor-mediated uptake, reversing the original decreased availability.

The functional deficiency of iron follows complexation of the host metal by the fiber surface and is the critical event that produces a biological effect after asbestos exposure. The host will respond with (1) increased expression of importers, (2) ferrireduction (i.e., an “oxidative burst”), (3) the activation of kinases/phosphatases and transcription factors, (4) the release of relevant mediators, and (5) both inflammation and fibrosis. Through the interactions between the cell and the fiber, new metal homeostasis will be determined, and this will include increased total iron concentrations with sufficient metal to meet cell requirements for continued function.

Following human inhalation, the greatest deposition of particles/fibers is in the 16th to 19th generations of airways, which includes the respiratory bronchioles [85,86,87,88]. There is a gravitational gradient in the ventilation distribution, with dependent regions of the lung (i.e., the lower lung fields) receiving more of each breath than the non-dependent regions. Particle/fiber deposition can often be greatest in the lower lung fields [89]. Accordingly, inflammation and fibrosis following asbestos exposure (e.g., respiratory bronchiolitis and respiratory bronchiolitis–interstitial lung disease) can present in the base of the lower respiratory tract. With increased exposure, inflammatory and fibrotic responses will be observed in the distal alveoli and interstitium (e.g., desquamative interstitial pneumonitis, non-specific interstitial pneumonitis, and usual interstitial pneumonitis). Pleural inflammation and fibrosis (e.g., pleural effusions and plaques, respectively) will follow lymphatic translocation of the asbestos to the membranes, enveloping the lungs through identical or similar mechanistic pathways.

## 5. The Formation of Ferruginous Bodies and Iron

In a primary pathway of defense in the lower respiratory tract, macrophages phagocytose particles/fibers and promote their mucociliary and lymphatic clearance [90,91,92]. This is efficient when the mean diameter of the particles/fibers is smaller than the mean diameter of the macrophage (10 microns) but, when the size exceeds this, there can be incomplete phagocytosis and clearance [93]. This produces an immobile macrophage, and the phagocytosed particles/fibers remain in the lung (unless they dissolve) [94,95,96]. An inability to clear asbestos from the lower respiratory tract initiates a host process of iron biomineralization (a process through which a living organism produces a mineral including an oxide) [97,98]. The fiber along with the mineral coat generated by the biomineralization response are collectively recognized as an asbestos body [98,99]. The precise composition of the coat is not known but has been claimed to possibly include iron, metals other than iron (calcium and magnesium), phosphate, protein, and mucopolysaccharides. As a result of the inclusion of iron, asbestos bodies are golden-brown in color. The coat along the fiber is frequently segmented into either spherical or rectangular units along the fiber, and therefore, can be beaded and segmented [100]. The distribution of the coat can be inhomogeneous, and the asbestos body can have a dumbbell or lancet appearance. The ends of the coated fibers can be knobbed, although sheath-like coats are also common [100]. Branched forms of asbestos bodies result from the deposition of a coat on a splayed fiber. Curved and circular asbestos bodies are also observed with very thin fibers (diameter of 0.2 micron or less).

Asbestos bodies result only after exposure to longer fibers (greater than 10–20 micron) since shorter fibers are phagocytosed and cleared via the mucociliary and lymphatic clearance pathways. Asbestos bodies are usually at least 20 microns but can be hundreds of microns long and have a diameter of less than 0.5 to 5 micron [19]. Most inhaled asbestos fibers do not have a ferruginous coat, with uncoated fibers accounting for 95–97% of the total [19]. Fibers less than 20 micron in length rarely become coated in the human lung, while virtually all fibers 80 microns or greater in length are coated. Thicker fibers are more likely to become coated than thinner fibers [101].

Since the number of asbestos bodies correlates with the burden of uncoated fibers 5 micron or longer, the vast majority isolated from the lungs of exposed workers demonstrate an amphibole (e.g., amosite or crocidolite) core [101], despite the bulk of commercial asbestos (95% or more) being chrysotile [101]. Chrysotile asbestos bodies do occur but account for a very small number of asbestos bodies (usually less than 1 to 3% with a mean of 0.14%) [100,101]. The infrequency of asbestos bodies following chrysotile exposure results from its shorter length relative to the commercial amphiboles, greater phagocytosis, dissolution by macrophages, and more efficient clearance from the lower respiratory tract [101].

In animal models, asbestos body formation can be observed within 2–3 months of exposure [101]. Asbestos body formation has been documented in lung tissue digests of infants from 3 to 12 months of age, suggesting that the time course for its formation is similar in humans [101]. Asbestos bodies can commonly be recovered from the lungs of the majority of adult humans without occupational exposure to any fibers, with 21% to 100% (median value of 90%) of lung tissues at autopsy demonstrating them [101]. Older individuals and smokers can have a greater number [101]. Asbestos bodies isolated from the lungs of those in the general population also have an amphibole core comparable to those isolated from asbestos workers.

The production of asbestos bodies occurs within macrophages on biopersistent fibers, which cannot be cleared from the lower respiratory tract (i.e., the amphiboles). They can be generated in vitro but cannot be produced in an acellular environment [102]. An analysis of their coat revealed the same elements in all asbestos bodies, with only minor differences in the relative amounts of the elements [100]. The component with the greatest concentration in the coat is iron. Asbestos bodies almost always provide a strong positive reaction with Perls’ Prussian blue stain, indicating the presence of iron. The iron content in the coat of asbestos bodies is frequently higher than that in the crystal lattice of any fiber. The iron in the coat can be explained by the metal originating from either (1) the crystal lattice and concentrating at the fiber surface after dissolution or (2) host metal pools. There are no data to support the source of iron being the fiber, and dissolution has not been delineated during the formation of asbestos bodies. Furthermore, iron will accumulate in a coat after exposure to fibers, as well as particles, with no or minimal iron in the lattice (e.g., anthophyllite, tremolite, and chrysotile) [103,104].

In silicates such as asbestos, isoelectric points support higher deprotonation of silanol groups (-Si-OH) relative to other functional groups (e.g., -Mg-OH), creating a heterogeneous electronegative charge along the surface. Iron, as well as other metals, will be complexed and accumulate at those sites along the surface populated by specific functional groups. The complexation of host cell iron by surface silanol groups initiates the formation of an asbestos body. The iron complexed by the fiber originates from low-molecular-weight pools of the metal (e.g., phosphates) rather than strongly coordinated metal (e.g., hemoglobin or transferrin) in host cells. For continued function and survival, the host cell requires the iron sequestered by the fiber (Figure 5).

In addition to importing more iron, the macrophage attempts to mobilize the complexed metal by producing superoxide (i.e., it generates an “oxidative burst”) at the phagosome/phagolysosome membrane. After reduction of the complexed Fe^3+^ to Fe^2+^ by O^2−^, the metal cation can be displaced from the asbestos surface.

Fe^2+^ concentration is a major determinant that controls iron biomineralization pathways. The generated Fe^2+^ is oxidized and can undergo hydrolysis in the cell environment, creating poorly crystalline iron oxyhydroxide: Fe^2+^ → Fe^3+^ → Fe(OH)^2+^→ Fe(OH)^2+^ → Fe(OH)_3_ (ferrihydrite). Analyses support that the coat on the asbestos bodies has a high iron concentration, that most of the metal is oxidized (i.e., ferric), and that its composition is similar to ferrihydrite [98,105,106,107]. Within a short period of time following asbestos exposure, this iron accumulates in the cytoplasm of the macrophages in close proximity to the fibers [108]. Macrophages incubated for 4 h or more will demonstrate uptake and positivity with Perls’ Prussian blue. Furthermore, ferrihydrite will rapidly precipitate from ferric compounds (e.g., FeCl_3_ and Fe_2_(SO_4_)_3_) in buffered solutions. There is an association between the iron-containing coat and the SiO_2_ layer of the asbestos since the latter provides the template for the initial complexation of the metal, and this is followed by its reduction, displacement, and precipitation as ferrihydrite [107,109]. This is comparable to iron-oxidizing bacteria, which utilize carbonaceous (e.g., lipopolysaccharide) fibers with a high negative charge as a template for biomineralization [110]. Increases in the concentrations of host iron result in higher rates of asbestos body formation since more metal is available for incorporation in the coat. Accordingly, welding and the mining of iron ore can be associated with elevations in metal concentrations and asbestos body formation in the lungs [99].

A series of reactions follow, and these involve interactions between competing Fe^2+^-induced biomineralization pathways. Due to its strong reducing capacity, Fe^2+^ further reacts with and provokes the conversion of ferrihydrite to secondary, more crystalline forms [111,112]. Subsequently, ferrihydrite in the coat can further react through hydrolysis–precipitation processes to produce goethite (Fe(OH)O), lepidocrocite (γ-FeO(OH)), magnetite, and hematite (Fe_2_O_3_) [113]. The phases vary in their precipitation extent, rate, and residence time, all of which are primarily a function of Fe^2+^ concentration; temperature and pH can also contribute [114,115]. The specific iron content of the coat (ferrihydrite, goethite, lepidocrocite, magnetite, and hematite) will vary, and this may correspond to the proximity to the fiber surface. However, analysis of the asbestos body demonstrates that the more crystalline forms of iron oxyhydroxides (e.g., hematite) are frequently either absent or present in amounts well below 5%, while ferrihydrite is the major component [98].

Iron oxyhydroxide nanoparticles that form at the fiber surface can participate in further adsorption of metals. As a result of a high surface area and intrinsic reactivity, ferrihydrite and other iron oxyhydroxides function as a “sink” for numerous metals in surface environments [116,117,118]. The retention mechanisms of metal cations and oxyanions on various iron oxyhydroxides can include (1) surface complexation, (2) surface precipitation, and (3) structural incorporation [119]. Hydroxyl groups on ferrihydrite control its chemical reactivity [120]. Elemental distribution maps acquired on asbestos bodies confirm that the surface is an efficient scavenger for metals [98]. Among those metals that can be bound through such a pathway, calcium has repeatedly been observed in the coat of the asbestos body [19,101]. Calcium is deposited on the fiber as either a phosphate or oxalate [101,121]. Reflecting the capacity of ferrihydrite and other iron oxyhydroxides for the sorption of metal cations, there is a correlation between calcium and iron in the coat of asbestos bodies. The amount of calcium and iron deposited on the fiber can range up to 1% and 10% (*w/w*), respectively, the content of the two metals can be directly proportional to each other, and they can co-localize in the coat [101,122]. Not all coats include both calcium and iron and, in humans and animal models, some will demonstrate only calcium phosphate or calcium oxalate crystals [20,101]. However, calcium phosphate and oxalate asbestos bodies are infrequent relative to those that are iron containing. Studies have similarly demonstrated that magnesium can sometimes participate in the process of asbestos body formation [107]. Furthermore, gadolinium and radium can be demonstrated in asbestos bodies following their use in radiographic imaging [123]. With the exception of silicon, all elements appear to co-localize to the coat on the asbestos body [100]. This likely reflects the capacity of iron oxides to bind these metals, leading to deposition/retention on the surface and the formation of asbestos bodies. Alternatively, a metal other than iron can be complexed by the functional groups at the fiber surface. This latter pathway can account for the rapidity of changes in the metal concentration [124].

Proteins and mucopolysaccharides have also been proposed as possible components of an asbestos body [98]. Comparable to metal cations, quaternary ammonium groups of a protein can be adsorbed onto negatively charged portions of a surface. Subsequently, the adsorption of positively charged proteins would localize to those segments of the fiber surface with greater silanol concentrations as a result of the negative charge. The inclusion of protein is anticipated to be indiscriminate and of little consequence to the organization of the asbestos body. In contrast to this, it has been proposed that the coat includes ferritin since the inorganic core of this storage protein can contain crystalline iron particles in the same size range [98,115]. An analytical comparison supported the idea of ferritin (or possibly hemosiderin) in an aggregated and/or misfolded form being a component of the asbestos body [98]. While its expression is increased in cells exposed to asbestos, ferritin does not stain with Perls’ Prussian blue [9,125,126,127,128]. In addition, immunohistochemistry does not confirm the presence of ferritin in the coat of asbestos bodies [45]. The formation of the crystalline material in an asbestos body has similarly been attributed to hemosiderin [19]. Hemosiderin is an iron storage protein that results from the incomplete degradation of ferritin, has a higher iron-to-protein ratio, is less soluble in aqueous solutions, and is considered a more stable and less available form of metal. However, the process of asbestos body formation occurs in the lysosome, which is normally the site of the degradation of both ferritin and hemosiderin to iron [129]. It is problematic to characterize either ferritin or hemosiderin in the surface coat with the formation of the asbestos body occurring at the cellular site of ferritinophagy.

Similar to asbestos, other particles/fibers form a ferruginous body, which comprises particles/fibers, and a coat, which includes iron [101,104]. This includes cigarette smoke particles; sheet silicates such as talc, mica, and kaolinite; diatomaceous earth; metal oxides; metals (with cores of titanium, iron, chromium, and aluminum); coal dust; woodstove dust; zeolites including erionite; silicon carbide; fiberglass; cotton dust; synthetic textile fibers; and human-made mineral fibers [101,104,130]. The coat in all these bodies contains iron, which stains positively with Perls’ Prussian blue. Similar to asbestos, these other particles/fibers function to complex iron [131]. It is proposed that the coats on these other particles/fibers reflect a disruption in cell iron homeostasis comparable to that following exposure to asbestos and result from the same process [100].

It has been suggested that the coat in the asbestos body is a protective mechanism. There is sequestration of the iron as oxyhydroxides, in which the valence sites of the metal are fully coordinated and unavailable for electron exchange (e.g., free radical generation). There is diminished generation of oxidative stress and reduced toxicity in coated fibers relative to uncoated fibers [20,132]. Iron in the coat of asbestos bodies does not appear to directly participate in the generation of reactive oxygen species [100]. The exposure of animals and humans to equivalent iron oxides (intranasal, intratracheal, intravenous, and inhaled) are most frequently without biological effects [133,134,135,136]. In medicine, iron oxide nanoparticles are widely used as therapeutic, delivery, and diagnostic agents.

Finally, in a pathway of iron biomineralization approximating asbestos body formation, several bacteria incorporate iron from their environment to synthesize intracellular nanoparticles of magnetite or greigite (Fe_3_S_4_) in organelles called magnetosomes [137,138]. This biomineralization also occurs within intracellular membranous vesicles that originate from invaginations of the cytoplasmic membrane [139]. The bacteria take up either a dissolved ferrous or ferric cation from the environment, store it in the cell as Fe^3+^, reduce it for trafficking to magnetosomes, and precipitate it as magnetite after the oxidation of Fe^2+^. The precursors include oxidized phases, including ferrihydrite. In another related pathway of iron biomineralization, iron plaque formation in plants is dependent on the availability of Fe^2+^, which reacts with oxygen to generate Fe^3+^ oxyhydroxides and deposits it on root surfaces [140,141]. Such iron plaques include crystalline iron oxyhydroxides and demonstrate positive staining with Perls’ Prussian blue [142,143]. On plant roots, iron plaque can function to bind other metals (e.g., cadmium, lead, zinc, and aluminum) impeding their entry [142]. These iron biomineralization pathways (magnetosomes and iron plaque) share features with asbestos bodies, including the reduction and oxidation of iron, precipitation of iron oxides, and adsorption of other metals.

## 6. Disparities between the Biological Impacts of Asbestos and Iron

There is no evidence that exposure to either amphibole or chrysotile asbestos is associated with the release of structural iron from the crystal lattice, which could have a biological effect. After exposure to the lower respiratory tract, the crystal lattice of amphiboles does not change and, as a result of its insolubility at any pH value, is biopersistent [19,96]. Attempts to clear amphiboles using mucociliary and lymphatic pathways are incomplete, resulting only in the translocation of some fibers to the mesothelial surface of the pleura. In contrast, macrophages dissociate the crystalline structure of chrysotile, leaving an unstable and amorphous silicate [20]. This process results in the thin rolled sheet structure of chrysotile breaking apart and decomposing into smaller fragments. The fragments are then successfully cleared from the lungs via mucociliary and lymphatic clearance. Most chrysotile is not biopersistent in the lungs [144,145,146,147,148,149,150,151].

In retained asbestos, surface functional groups on asbestos initiate the complexation of host iron sources. The biological effect of asbestos is dependent on this sequestration of iron by the fiber surface. The greater clearance of chrysotile will diminish the biological effect [144,145,146,147,148,149,150,151]. In addition, the number of functional groups at the asbestos surface will ultimately be dictated by the lattice composition. Therefore, the higher percentage SiO_2_ in the lattice of amphibole asbestos can result in higher surface silanol numbers, a larger capacity to complex iron and disrupt metal homeostasis, and greater biological effects relative to chrysotile.

## Figures and Tables

**Figure 1 ijms-24-12390-f001:**
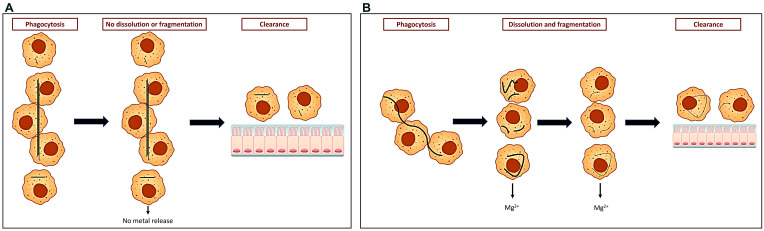
Clearance of asbestos from the lower respiratory tract. Exposure to longer amphibole fibers can result in “frustrated phagocytosis” and failure of clearance pathways, while shorter fibers and fragments can be successfully phagocytosed and removed via the mucociliary pathway. Phagosomal degradation of amphibole with release of metals does not occur (**A**). In contrast, longer chrysotile fibers are phagocytosed and the brucite layer is destroyed in an acidic environment with release of metal (e.g., magnesium) constituents. This results in fragments that can eliminated from the lower respiratory tract via the mucociliary clearance pathway (**B**).

**Figure 2 ijms-24-12390-f002:**
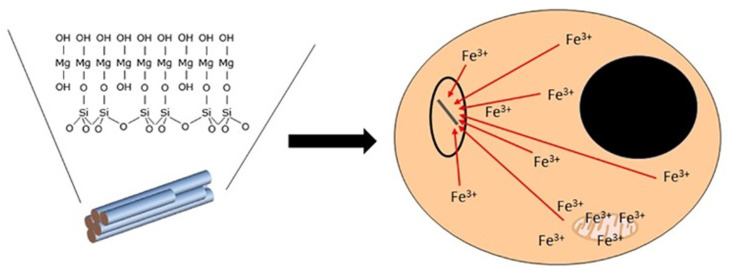
Cell exposure to asbestos and iron homeostasis. Functional groups (e.g., -Si-O^−^ groups) on the surface of the fibrous silica will complex host sources of intracellular iron. There is a loss of metal required for critical functions in the cell. Mitochondria are especially vulnerable as processes in this organelle demonstrate significant dependence on the availability of this metal (e.g., Krebs cycle and electron transport system).

**Figure 3 ijms-24-12390-f003:**
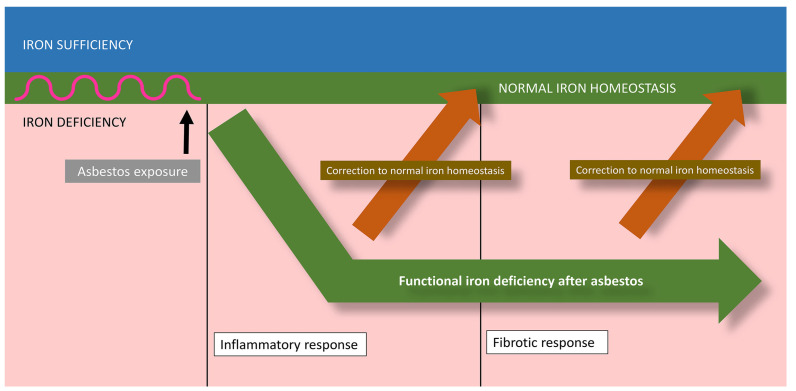
Exposure to asbestos causes a functional iron deficiency. The host response reflects an attempt to return to normal iron homeostasis with metal available for critical functions and cell survival. This includes inflammation and fibrosis.

**Figure 4 ijms-24-12390-f004:**
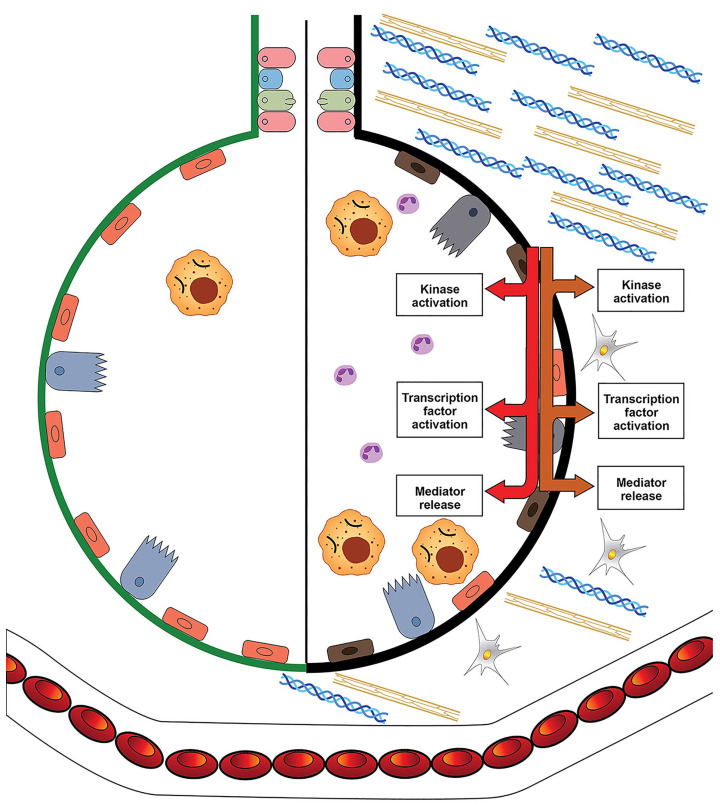
The biological effects of asbestos and iron. Decreased iron availability affects kinase and transcription factor activation, which coordinate a release of mediators relevant to inflammatory and fibrotic responses. An influx of inflammatory cells (e.g., macrophages and neutrophils) corresponds with the decreased metal availability. There is also an increased number of fibroblasts and a deposition of collagen (represented by blue helical units), elastin (represented by yellow units), and extracellular polymeric substances, which correspond to functional iron deficiency.

**Figure 5 ijms-24-12390-f005:**
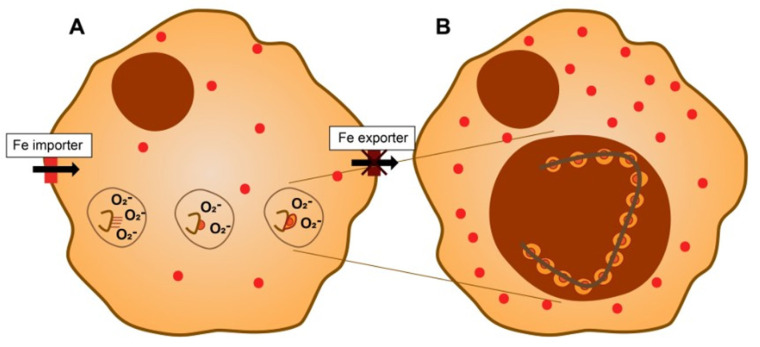
Asbestos body development and iron. In the phagolysosome, asbestos complexes cell iron, impacting functional iron deficiency (**A**). The host cell responds with increases in the import and decreases in the export of iron, elevating storage of the metal in ferritin (represented by red dots). In addition, NADPH oxidoreductases at the phagosome membrane will increase the generation of superoxide to reduce Fe^3+^ to Fe^2+^, which is released by surface functional groups. The Fe^2+^ reacts with oxygen to form iron to form oxyhydroxides including ferrihydrite. Nanoparticulate iron oxyhydroxides provide a large surface, which can react with additional metals (e.g., iron, calcium, magnesium, gadolinium, and radium). The iron-containing structure that results corresponds to the template provided by the position of the functional groups (homogeneous, segmental, barbell, etc.) at the asbestos surface (**B**).

**Table 1 ijms-24-12390-t001:** Ideal molecular formulas of fibrous silicates included in asbestos.

Actinolite	Ca_2_(Mg,Fe^2+^)_5_Si_8_O_22_(OH)_2_
Amosite	(Mg,Fe^2+^)_7_Si_8_O_22_(OH)_2_
Anthophyllite	Mg_7_Si_8_O_22_(OH)_2_
Crocidolite	Na_2_Fe_3_^2+^Fe_2_^3+^Si_8_O_22_(OH)_2_
Tremolite	Ca_2_Mg_5_Si_8_O_22_(OH)_2_
Chrysotile	Mg_3_Si_2_O_5_(OH)_4_

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
