# Peer review of "Asbestos and Iron"

_ijms, 2023, doi:10.3390/ijms241512390_

Round 1

Reviewer 1 Report

Manuscript ID: ijms-2514248 – Reviewer’s comments

General comment

This review from Ghio and colleagues focuses on the partecipation of iron in the pathogenesis of asbestos related diseases. In their introduction the authors, who represent excellence in iron related asbestos toxichology research, state that “theories of disease pathogenesis  following asbestos exposure were developed which focused on a participation of iron” … one suggesting “that pathogenesis of asbestos-related disease is dependent on the complexation chemistry of iron” and the other on its “oxidant chemistry” maily through “an in vivo production of  hydroxyl radical by iron”. Anyway in this review they pursue only the first theory, in the light of which they starts from a detailed descriprion of iron chemistry and reactivity of asbestos fibers, subsequently the consequences of asbestos internalization by macropghages (only) on cellular iron homeostasis are reported and finally the formation of asbestos bodies is discussed.

In the reviewer opinion the two theories are not mutually exclusive and should both be discussed. Aternatively the title of the review should be reconsidered, as for example: "Role of asbestos-induced cellular iron deficiency in the pathogenesis of asbestos related diseases".

Furthermore the perspective supported by Ghio and colleagues in the present review is quite different from that in 2004 (Ghio et al. DOI: 10.1080/01926230490885733). Quoted verbatim: “IMPLICATIONS Free radical generation by the fiber and particle is mediated in some part by coordinated metal. Ferruginous bodies represent an attempt by the host to sequester the metal adsorbed to the surface of a fiber and diminish the oxidative challenge presented by a fiber or particle. While much of the iron coordinated onto the surface will be stored in a catalytically less reactive state within ferritin included in the ferruginous body, some portion of this metal will ultimately be catalytically active and therefore capable of supporting the generation of an oxidative stress. Consequently, the observation of ferruginous bodies on microscopic inspection of lung tissue should be interpreted as supporting a specific mechanism of injury (i.e., a metal-catalyzed oxidative stress). This would apply not only to tissue injury after exposure to fibers and particles but also is pertinent to damage associated with surfaces of many appliances placed in the body if that surface presents oxygen-containing functional groups (Jordan et al., 2002)”.

This change of view may have a scientific basis, which the reviewer thinks should be justified.

The reviewer also found some shortcomings and wrong attribution (later discussed in the specific comments) regarding the references of some statements. In general, the reviewer found this review too focused on the theory currently supported by the authors, so as not to maintain the expectations generated by the title.

In this form and without discussing the issues raised by the reviewer this review is not suitable for publication, unless the subject dealt with is limited to the discussion of the authors theory about the role played by iron sequestration in the induction of asbestos-related diseases.

Specific comments

Page 1, line 57. Iron involvement in the biological effects of asbestos are also supported by some genetic studies investigating the role of iron related gene polymorphisms in the etiology of asbestos related cancers (mesothelioma, MPM and lung cancer, LC). These studies report three SNPs, localized in the ferritin heavy polypeptide, transferrin, and hephaestin genes,  significantly associated with protection against development of MPM and LC (DOI: 10.1080/15287394.2015.1123452 and  doi: 10.1080/15287394.2019.1694612). Iron signature in asbestos-related diseas should be also discussed.

Page 5, figure 2. The process by which the iron would enter the phaglo-lysosome (through which carrier?) should be better explained.

Page 6, line 182. The authors describe only the increase in iron import, but the cells could also limit the export of iron through the ferroportin/hephaestin (or ceruloplasmin) system, as occurs (thanks to hepcidin activity) during inflammation. A variant of hephestin has been associated with a lower risk of developing related asbestos neoplasms. The authors should discuss this point (iron export) also at Page 7 line 256 and figure 5.

Page 6 line 200. “The electron transport chain in the mitochondria can be a major source of superoxide production (46, 47)” perhaps this statement is true for most cells, anyway in phago cytes the major source of superoxide is the oxidative-burst. In this context the reviewer believes that the role of neutrophils should also be considered, as they engulf asbestos fibers (personal communication), produce more superoxide than macrophages, release, upon activation,  lactoferrin and elastase, which the authors argue may release the iron bound to elastase (page 7, lines 246-248). In figure 4 these cells are represented within the alveolus, but their role is not dissected. Furthermore the effects of asbestos internalization on iron homeostasis of mesothelial cells (which display phagocytic function DOI: 10.1097/IGC.0000000000000697 and differentiate into macrophage-like cells DOI: 10.1111/j.1600-0463.2011.02803.x) shluold be also discussed since the pathogenesis of chrysotile-induced mesothelial carcinogenesis seems closely associated with local iron overload (doi: 10.1002/path.4075; DOI: 10.1038/bjc.1989.344).

Page 7 line 215-232. “Inflammation is included in the host response to a functional iron deficiency”. This statement is debatable and should be riconsidered, since inflammation itself (via hepcidin), induces systemic iron deficency and ferritin is an acute phase protein. The role of iron deficiency in the induction of inflammation is mostly supported by indirect studies, when phagocytosis of the fibers by the phagocytes itself induces the release of inflammatory mediators. Cellular iron deficiency may help to explain the inflammation which generate a pro-inflammatory microenvironment favoring cancerogenesis, anyway radical generation (presumably via catalytic action by the Fenton reaction induced by iron complexation by the fiber surface) is necessary for neoplastic transformation to occur, and authors should discuss also this issue.

Page 8,line 174: The formation of ferruginous bodies (FB)

General comment: the authors  disregard two key points in the biology of FB.

(i)                 First the by now acquired certainty that asbestos fibers can absorbe many different proteins for example in: https://doi.org/10.1111/j.1349-7006.2011.02087.x; https://doi.org/10.3390/ijerph15010104). Ferritin, one of the protein absorbed by asbestos (10.1080/00984109708984069) is presently recognized as a major component of FB (https://doi.org/10.1080/15287390701380906; DOI: 10.1038/srep44862) together with a minor amount of free iron. So, showing the presence of ferritin in FB is not problematic (page 11, line 434), it is a fact.

Ferritin plays a key role in FB formation, where  this protein is absorbed by the fibers (10.1080/15287394.2022.2164391),  is continuously synthetized and  is also secreted in extracellular vesicles. Indeed asbestos exposed alveolar macrophages are characterized by a high turnover due to the high level of ferroptotic cell death. Upon ferroptosis macrophages release extracellular vesicles containing Fe-loaded ferritin (doi: 10.15430/JCP.2021.26.4.244). The authors themselves postulated for the first time an alteration of lung iron homeostasis after asbestos exposure relying on elevated ferritin levels in bronchoalveolar lavage of exposed individuals (doi: 10.1089/ars.2007.1909).

Furthermore, recently Zangari et al (https://doi.org/10.1080/15287394.2022.2164391), have shown that asbestos fiber exert themselves a iron (II) oxidative activity. Since the iron loading process in the ferritin cage is complex (https://doi.org/10.15430/JCP.2021.26.4.244 ) and specific proteins are required for transporting iron (II) to ferritin cage for oxidation and loading. Therefore the iron oxidative activity of the fibers may compete with ferritin loading, favours free iron binding on the fiber surface and inhibit ferritin derived iron availability (increasing iron deficiency) in a context of iron overload.

In the reviewer opinion the sequestration of ferritin by fibers may represent even a further element contributing to the change of iron homeostasis, and cannot be forgotten. Even if they are quoted, the findings of other authors that the synthesis of ferritin is increased in different cell types exposed to asbestos should also been deeply discussed.

(ii)               The second point regards the role plaid by FB in the asbestos-related diseases. Is it a defence mechanism? Or can it contribute to the cell/tissue damage? Some authors have provided evidence that asbestos bodies really can propagate damage and stimulate the inflammatory process, even if, from some points of view can inhibit the cytotoxic power of these peculiar structures. I think that the latter possibility deserves to be discussed. Some researchers  presented evidence that FB can contribute to mantain the toxicity of AB and their stimulus to chronic inflammation. They can damage DNA (https://doi.org/10.1080/15287394.2023.2181899; https://doi.org/10.1080/15287394.2012.688478). Even if these findings are in some cases debatable (DNA damage), and  wait to be confirmed in “in vivo” models, they must be mentioned and discussed in a complete review.

Page 9, Figure 5. Please indicate which Fe exporter would be blocked. Please see also the previous comment on hephaestine/ferroportin-mediated iron export.

In the reviewer opinion the quality of the images is quite poor and they add nothing to the text, it is recommended to improve them or remove them altogether.

Page 10, line 389. Please note that recently Bardelli and colleagues (DOI: 10.1007/s10653-023-01557-0) reported in the FB coating “the presence of ferrihydrite, and, to a lower extent, of goethite, as the major phases, and the absence of hematite”. This reference should be added.

Page 12, References

Reference 101: in the present review, the author mention that : iron in the coat of AB does not appear to directly participate in the generation of reactive oxygen species (ref 101).  In that paper Bardelli et al. don’t  assert this. Rather they   say: “….. indicated that hematite and metallic iron, whose presence in the AB was claimed in a previous study, are absent or present in amounts well below 5%, and thus that the AB are mainly composed by ferritin and/or hemosiderin”.

Author Response

Manuscript ID: ijms-2514248 – Reviewer 1

Reviewer’s comment: This review from Ghio and colleagues focuses on the participation of iron in the pathogenesis of asbestos related diseases. In their introduction the authors, who represent excellence in iron related asbestos toxicology research, state that “theories of disease pathogenesis  following asbestos exposure were developed which focused on a participation of iron” … one suggesting “that pathogenesis of asbestos-related disease is dependent on the complexation chemistry of iron” and the other on its “oxidant chemistry” mainly through “an in vivo production of  hydroxyl radical by iron”. Anyway in this review they pursue only the first theory, in the light of which they start from a detailed description of iron chemistry and reactivity of asbestos fibers, subsequently the consequences of asbestos internalization by macrophages (only) on cellular iron homeostasis are reported and finally the formation of asbestos bodies is discussed.

In the reviewer opinion the two theories are not mutually exclusive and should both be discussed. Alternatively the title of the review should be reconsidered, as for example: "Role of asbestos-induced cellular iron deficiency in the pathogenesis of asbestos related diseases".

Furthermore the perspective supported by Ghio and colleagues in the present review is quite different from that in 2004 (Ghio et al. DOI: 10.1080/01926230490885733). Quoted verbatim: “IMPLICATIONS Free radical generation by the fiber and particle is mediated in some part by coordinated metal. Ferruginous bodies represent an attempt by the host to sequester the metal adsorbed to the surface of a fiber and diminish the oxidative challenge presented by a fiber or particle. While much of the iron coordinated onto the surface will be stored in a catalytically less reactive state within ferritin included in the ferruginous body, some portion of this metal will ultimately be catalytically active and therefore capable of supporting the generation of an oxidative stress. Consequently, the observation of ferruginous bodies on microscopic inspection of lung tissue should be interpreted as supporting a specific mechanism of injury (i.e., a metal-catalyzed oxidative stress). This would apply not only to tissue injury after exposure to fibers and particles but also is pertinent to damage associated with surfaces of many appliances placed in the body if that surface presents oxygen-containing functional groups (Jordan et al., 2002)”.

This change of view may have a scientific basis, which the reviewer thinks should be justified.

Response to reviewer’s comments: The reviewer recommends that the current manuscript also include focus to the “oxidant chemistry” of iron and this can involve “an in vivo production of  hydroxyl radical by iron”. Several decades of science (two since the perspective quoted by the reviewer: Ghio AJ, Churg A, Roggli VL. Ferruginous bodies: implications in the mechanism of fiber and particle toxicity. Toxicol Pathol 2004; 32(6):643-9. doi: 10.1080/01926230490885733) support an involvement of complexation chemistry of iron in the biological effects of asbestos. This involvement of the complexation chemistry of iron in the biological effect of asbestos is addressed throughout the entirety of this submitted manuscript. However, regarding inclusion of the oxidative chemistry of iron in the biological effects of particles and fibers (e.g. asbestos), the authors have more recently stated that:

“Life is ferrocentric with iron being an essential micronutrient required by every cell. A favorable oxidation-reduction potential and a relative abundance led to its evolutionary selection for a wide range of fundamental functions. Following the introduction of oxygen to the atmosphere as a product of photosynthesis, water-soluble ferrous ion (Fe2+) was effectively removed from the Earth’s crust. The resultant ferric ion (Fe3+) remained but, being insoluble in water, at concentrations inadequate to meet the requirements for life; the concentrations of Fe3+ in water at physiologic pH values approximate 10−18 M while those required for life approach 10−6 M. Accordingly, greater quantities of metal had to be procured to support emerging life. This challenge was realized by living systems by utilizing two major pathways to acquire essential iron: 1) the chemical reduction of Fe3+ to Fe2+ (i.e. ferrireduction) with its subsequent import and utilization and 2) the complexation of Fe3+ with chelators coupled with receptors for uptake of the complex and employment of the metal. In addition to solubility limiting its availability, iron-catalyzed generation of radicals presented a potential for oxidative stress; improperly sequestered iron has a theoretical potential to catalyze toxic reactive oxygen species. Such reactivity mandated that iron homeostasis be tightly controlled. Living systems evolved strategies to regulate the procurement of adequate iron for cellular function and homeostasis precluding damage to biological macromolecules. The import, storage, and efflux of this metal is vigilantly regulated. Accordingly, life exists at the interface between iron-deficiency and iron-sufficiency.” (Ghio AJ, Soukup JM, Dailey LA, Madden MC. Air pollutants disrupt iron homeostasis to impact oxidant generation, biological effects, and tissue injury. Free Radic Biol Med. 2020;151:38-55. doi: 10.1016/j.freeradbiomed.2020.02.007.)

Life is positioned at an interface between iron deficiency and sufficiency. This is evident with a significant proportion of the human population defined to be iron deficient (e.g., toddlers and children of preschool age internationally have a rate of iron deficiency anemia that can approximate 50%). Iron overload (focal or otherwise) and a possible support of an uncontrolled production of hydroxyl radical is extremely rare in any living system, including humans, if it does occur. The control of the potential for iron to support such oxidant generation can reflect design. The intracellular concentration of iron can approximate 1 to 10 µM (this is total). An empty or labile coordination site on the iron cation is required to support electron exchange (e.g., hydroxyl radical generation). The intracellular concentration of compounds and substances that complex and assume empty or labile coordination sites on iron (e.g. phosphates and carboxylates) can be millions of times higher than this. Under these conditions, an empty or labile coordination site is precluded and conditions which might support generation for hydroxyl radical do not exist (again, this is in a living system). Subsequently, with this background and the data presented in the submitted manuscript, the authors cannot fairly address the “oxidant chemistry” of iron in the biological effect of asbestos. Much of this data has been made available in the last few decades.

The association between asbestos and iron is comparable to disease in many fields of medicine (Beavers CJ, Ambrosy AP, Butler J, Davidson BT, Gale SE, Piña IL, Mastoris I, Reza N, Mentz RJ, Lewis GD. Iron Deficiency in Heart Failure: A Scientific Statement from the Heart Failure Society of America. J Card Fail. 2023; 29(7):1059-1077. doi: 10.1016/j.cardfail.2023.03.025.)

Reviewer’s comment: The reviewer also found some shortcomings and wrong attribution (later discussed in the specific comments) regarding the references of some statements. In general, the reviewer found this review too focused on the theory currently supported by the authors, so as not to maintain the expectations generated by the title.

In this form and without discussing the issues raised by the reviewer this review is not suitable for publication, unless the subject dealt with is limited to the discussion of the authors theory about the role played by iron sequestration in the induction of asbestos-related diseases.

Response to reviewer’s comments: Based on these specific comments provided by the reviewer, the manuscript has been amended to include (TITLE):

“Asbestos and the complexation chemistry of iron”

Reviewer’s comment: Specific comments. Page 1, line 57. Iron involvement in the biological effects of asbestos is also supported by some genetic studies investigating the role of iron related gene polymorphisms in the etiology of asbestos related cancers (mesothelioma, MPM and lung cancer, LC). These studies report three SNPs, localized in the ferritin heavy polypeptide, transferrin, and hephaestin genes,  significantly associated with protection against development of MPM and LC (DOI: 10.1080/15287394.2015.1123452 and  doi: 10.1080/15287394.2019.1694612). Iron signature in asbestos-related disease should be also discussed.

Response to reviewer’s comments: The manuscript has been amended to include (INTRODUCTION):

“There is support for metal involvement in the biological effect of asbestos with inclusion of iron-related proteins in the genetic signatures of asbestos-related malignancies and both chelation therapy and phlebotomy demonstrating possible preventive effects against asbestos-induced carcinogenesis in animal models (5-9).”

The following references have been added (REFERENCES):

Jiang L, Akatsuka S, Nagai H, Chew S-H, Ohara H, Okazaki Y et al. Iron overload signature in chrysotile-induced malignant mesothelioma. J Pathol 2012 Nov;228(3):366-77. doi: 10.1002/path.4075.

Crovella S, Bianco AM, Vuch J, Zupin L, Moura RR, Trevisan E et al. Iron signature in asbestos-induced malignant pleural mesothelioma: A population-based autopsy study. J Toxicol Environ Health A 2016;79(3):129-41. doi: 10.1080/15287394.2015.1123452.

Celsi F, Crovella S, Moura RR, Schneider M, Vita F, Finotto L, et al. Pleural mesothelioma and lung cancer: the role of asbestos exposure and genetic variants in selected iron metabolism and inflammation genes. J Toxicol Environ Health A. 2019;82(20):1088-1102. doi: 10.1080/15287394.2019.1694612.

Reviewer’s comment: Page 5, figure 2. The process by which the iron would enter the phago-lysosome (through which carrier?) should be better explained.

Response to reviewer’s comments: With exposure to asbestos, the fiber is phagocytosed by macrophages. Iron is complexed by the fiber. This must occur intracellularly with the asbestos in the phago-lysosome since 1) asbestos in macrophages is localized to phago-lysosomes and 2) there is no extracellular iron available to be complexed by the fiber.

However, the manuscript does not specify how the metal is complexed to the asbestos surface in the phago-lysosome. The required investigation has not been accomplished to define this. With normal import of iron utilizing transferrin-Fe3+, there is acidification of the lysosome, release of the Fe3+, reduction of the Fe3+ to Fe2+ by STEAP, and movement of the Fe2+ across the membrane by DMT1. It is also recognized that NRAMP1 can facilitate transport of iron in the macrophage (Soe-Lin S, Apte SS, Mikhael MR, Kayembe LK, Nie G, Ponka P. Both Nramp1 and DMT1 are necessary for efficient macrophage iron recycling. Exp Hematol. 2010; 38(8):609-17. doi: 10.1016/j.exphem.2010.04.003). However, both DMT1 and NRAMP1 can transport the metal out of the phago-lysosome.

It is possible that functional groups at the surface of the fiber complex available metal which is in the immediate environs. A cascade effect would follow with those molecules/substances/organelles which lost metal then complexing adjacent sources of metal. Accordingly, metal accumulates on the asbestos surface and in the phago-lysosome and this would be associated with the observed deficiency in the cell and mitochondrial dysfunction (reflecting the loss of iron).

Reviewer’s comment: Page 6, line 182. The authors describe only the increase in iron import, but the cells could also limit the export of iron through the ferroportin/Hephaestin (or ceruloplasmin) system, as occurs (thanks to hepcidin activity) during inflammation. A variant of Hephaestin has been associated with a lower risk of developing related asbestos neoplasms. The authors should discuss this point (iron export) also at Page 7 line 256 and figure 5.

Response to reviewer’s comments: There is a scarcity of investigation regarding cellular export of iron. Until recently, the single iron exporter recognized has been ferroportin-1 (FPN1) whose activity is coupled with hephaestin or ceruloplasmin oxidation. Only in the last few years has extracellular vesicles with inclusion of ferritin been considered as an additional potential pathway of iron export.

Regarding ferroportin and asbestos, there have been only three articles that we can find. Investigation which has included hephaestin and asbestos is available in only one article (mentioned previously in this response and now referenced in the revised manuscript). It is difficult to provide much discussion on potential iron export after asbestos exposure based on this limited investigation.

The manuscript has been amended to include (“ASBESTOS, CELL IRON HOMEOSTASIS, AND BIOLOGICAL EFFECT”):

“Changes in the expression and activity of the metal exporter ferroportin-1 (FPN1) as well as exported iron concentrations have yet to be fully delineated but are also possible (Ghio A, Tan RJ, Ghio K, Fattman CL, Oury TD. Iron accumulation and expression of iron-related proteins following murine exposure to crocidolite. J Environ Pathol Toxicol Oncol. 2009;28(2):153-62 and Ghio AJ, Pavlisko EN, Roggli VL. Iron and Iron-Related Proteins in Asbestosis. J Environ Pathol Toxicol Oncol 2015;34(4):277-85.)”

Reviewer’s comment: Page 6 line 200. “The electron transport chain in the mitochondria can be a major source of superoxide production (46, 47)” perhaps this statement is true for most cells, anyway in phagocytes the major source of superoxide is the oxidative-burst.

Response to reviewer’s comments: The manuscript has been amended to read (ASBESTOS, CELL IRON HOMEOSTASIS, AND BIOLOGICAL EFFECT): “The host also attempts to re-acquire its metal directly sequestered by the fiber. Movement of iron necessitates ferrireduction to chemically reduce the metal to the ferrous state (e.g., transmembrane transport). This chemical reduction is frequently accomplished by superoxide generated in the cell (e.g., membranes of the cell, mitochondria, and lysosomes). The major source of superoxide after particle exposure is the phagocyte-associated NADPH oxidoreductase (Ghio AJ, Soukup JM, Stonehuerner J, Tong H, Richards J, Gilmour MI, Madden MC, Shen Z, Kantrow SP. Quartz Disrupts Iron Homeostasis in Alveolar Macrophages To Impact a Pro-Inflammatory Effect. Chem Res Toxicol 2019; 32(9):1737-1747. doi: 10.1021/acs.chemrestox.8b00301). An increased availability of iron (i.e. cell pretreatment with FAC) diminishes a functional iron deficiency and superoxide production following asbestos exposure (41). The “oxidative stress” after asbestos exposure includes mitochondria as a source since rotenone inhibits some portion of superoxide generation and iron uptake by the cell. The electron transport chain in the mitochondria can be an additional source of superoxide production (46, 47). Through a comparable pathway, cellular superoxide generation can increase after exposure to iron chelators (48, 49). The generated superoxide functions in ferrireduction with cell acquisition, transport, and translocation of iron. Such ferrireduction is an essential, and frequently limiting, reaction in iron acquisition, transport, and translocation (50, 51).”

Reviewer’s comment: In this context the reviewer believes that the role of neutrophils should also be considered, as they engulf asbestos fibers (personal communication), produce more superoxide than macrophages, release, upon activation,  lactoferrin and elastase, which the authors argue may release the iron bound to elastase (page 7, lines 246-248). In figure 4 these cells are represented within the alveolus, but their role is not dissected.

Response to reviewer’s comments: To fairly address the integration of the roles of neutrophils and macrophages in impacting iron homeostasis after exposures to particles (including fibrous particles such as asbestos) and infectious agents necessitates its own article. Numerous products of neutrophils and macrophages are focused on a mobilization of the metal (e.g., proteases), reduction of the metal (e.g., superoxide and its related products), and translocation back to the host (e.g., lactoferrin). A complete discussion of these multiple pathways is not possible here.

The manuscript has been amended to include (FIGURE 4 LEGEND): “The biological effect of asbestos and iron. Decreased iron availability affects kinase and transcription factor activation which coordinate a release of mediators relevant to inflammatory and fibrotic responses. An influx of inflammatory cells (e.g., macrophages and neutrophils) corresponds with the decreased metal availability. Products from the phagocytic cells can impact iron homeostasis with mobilization (e.g., proteases), ferrireduction (e.g., superoxide), and translocation (e.g., lactoferrin) of the metal allowing the host to restrict its availability. There is also an increased number of fibroblasts and a deposition of collagen (designated by blue helical units), elastin (designated by yellow units), and extracellular polymeric substances which correspond to the functional iron deficiency.”

Reviewer’s comment: Furthermore the effects of asbestos internalization on iron homeostasis of mesothelial cells (which display phagocytic function DOI: 10.1097/IGC.0000000000000697 and differentiate into macrophage-like cells DOI: 10.1111/j.1600-0463.2011.02803.x) should be also discussed since the pathogenesis of chrysotile-induced mesothelial carcinogenesis seems closely associated with local iron overload (doi: 10.1002/path.4075; DOI: 10.1038/bjc.1989.344).

Response to reviewer’s comments: The manuscript has been included to include this (ASBESTOS, CELL IRON HOMEOSTASIS, AND BIOLOGICAL EFFECT).

“Pleural inflammation and fibrosis (e.g., pleural effusions and plaques respectively) will follow lymphatic translocation of the asbestos to the membranes enveloping the lungs through identical or similar mechanistic pathways. However, in addition to macrophages and neutrophils, it is possible that mesothelial cells can contribute to the biological effect after differentiation to phagocytic cells and anti-gen presenting cells.”

The following references have been added (REFERENCES):

“Katz S, Balogh P, Kiss AL. Mesothelial cells can detach from the mesentery and differentiate into macrophage-like cells. APMIS  2011; 119(11):782-93. doi: 10.1111/j.1600-0463.2011.02803.x

Shaw TJ, Zhang XY, Huo Z, Robertson D, Lovell PA, Dalgleish AG, Barton DP. Human Peritoneal Mesothelial Cells Display Phagocytic and Antigen-Presenting Functions to Contribute to Intraperitoneal Immunity. Int J Gynecol Cancer 2016 Jun;26(5):833-8. doi: 10.1097/IGC.0000000000000697”

The last two references provided by the reviewer are not added to the manuscript as they are considered by the authors to misinterpret the exposure to an iron chelator as an “overload”.

Reviewer’s comment: Page 7 line 215-232. “Inflammation is included in the host response to a functional iron deficiency”. This statement is debatable and should be reconsidered, since inflammation itself (via hepcidin), induces systemic iron deficiency and ferritin is an acute phase protein. The role of iron deficiency in the induction of inflammation is mostly supported by indirect studies, when phagocytosis of the fibers by the phagocytes itself induces the release of inflammatory mediators.

Response to reviewer’s comments: The article carefully expresses a relationship of inflammation with asbestos and iron availability:

“Inflammation is included in the host response to a functional iron deficiency. The complexation of host iron by asbestos initiates pathways which culminate in a release of inflammatory mediators and inflammation. Cell iron deficiency following exposures to fibers activates kinases and transcription factors which are associated with a release of inflammatory mediators. Cell exposure to asbestos activates mitogen-activated protein (MAP) kinases and some portion of this response can be diminished by increasing the concentration of available iron (52-54). Transcription factors involved in expression of inflammatory mediators are also activated by particle/fiber exposure (55-58). Many of these same transcription factors control cell death and if the response to correct the functional iron deficiency is insufficient, some form of apoptosis will ensue (59-62). Comparable to MAP kinases, increased iron availability decreases the activation of transcription factors following exposures to particle/fiber (53, 54, 63). After impacting cell signaling and transcription factors, asbestos exposure will produce changes in expression of inflammatory mediators. Changes in protein expression for IL-6 and IL-8 after asbestos exposure are diminished by cell pretreatment with iron reflecting their association with metal availability (64). It is the functional iron deficiency after asbestos exposure which activates kinase signaling and transcription, increases release of mediators, and coordinates the inflammatory response.”

It is unclear what the reviewer is referring to as “debatable”. The references support the relationship between the cell and molecular basis for inflammation and iron availability. Changes in mediators (e.g., hepcidin), decreased serum iron levels, and anemias of chronic disease are products of the cell and molecular events. This is comparable to numerous other compounds and substances which complex iron to decrease availability of the metal and, like asbestos, are associated with inflammation, fibrosis, and neoplasm (e.g. other particles and fibers, microbials, bleomycin and doxorubicin). More references can be provided specifically on the relationship between inflammation and iron availability if greater support is thought necessary.

Reviewer’s comment: Cellular iron deficiency may help to explain the inflammation which generate a pro-inflammatory microenvironment favoring cancer genesis, anyway radical generation (presumably via catalytic action by the Fenton reaction induced by iron complexation by the fiber surface) is necessary for neoplastic transformation to occur, and authors should also discuss this issue.

Response to reviewer’s comments: This theory of inflammation and carcinogenesis is unsupported by several decades of investigation. The involvement of hydroxyl radical in the biological effects of asbestos is difficult to postulate. This requires empty or labile coordination sites to support electron transport. Iron in a living system is tightly complexed and carefully handled/transported. In a living system, the ratio of compounds and substances which complex iron to the metal absolutely precludes a generation of hydroxyl radical via the Fenton reaction. In any cell, the ratio of phosphates and carboxylates alone is 106 greater than iron.

The manuscript has been amended to include a discussion of the relationships between asbestos, iron, and carcinogenesis (ASBESTOS, CELL IRON HOMEOSTASIS, AND BIOLOGICAL EFFECT):

“Finally, malignancy is a biological effect of asbestos. A diverse group of cells in a tissue, each with its individual requirements for iron and pathways for metal acquisition, has access to a pool of iron. Comparable to exposures to numerous compounds and substances associated with carcinogenesis (e.g. cigarette smoke particle, erionite, and metals), asbestos is associated with a disruption in homeostasis and a functional deficiency of iron. In an environment in which iron is a limited resource, competition will arise between the cells for the metal. The functional iron deficiency after exposure to asbestos can apply a selective pressure on cell communities. Pathways for iron acquisition employed by a malignant cell allow it to compete more successfully for this nutrient which is essential for proliferation and growth (e.g., increased expression of iron importers such as transferrin and lactate and their respective receptors). A malignant cell subsequently will surpass a normal cell in the capacity to proliferate and grow in the iron-depleted environment. By competing more effectively for requisite iron, malignant cells will displace others and dominate the cell population.”

Reviewer’s comment: Page 8,line 174: The formation of ferruginous bodies (FB)

General comment: the authors  disregard two key points in the biology of FB.

(i)                 First the by now acquired certainty that asbestos fibers can absorb many different proteins for example in: https://doi.org/10.1111/j.1349-7006.2011.02087.x; https://doi.org/10.3390/ijerph15010104). Ferritin, one of the protein absorbed by asbestos (10.1080/00984109708984069) is presently recognized as a major component of FB (https://doi.org/10.1080/15287390701380906; DOI: 10.1038/srep44862) together with a minor amount of free iron. So, showing the presence of ferritin in FB is not problematic (page 11, line 434), it is a fact.

Ferritin plays a key role in FB formation, where  this protein is absorbed by the fibers (10.1080/15287394.2022.2164391),  is continuously synthetized and  is also secreted in extracellular vesicles. Indeed asbestos exposed alveolar macrophages are characterized by a high turnover due to the high level of ferroptotic cell death. Upon ferroptosis macrophages release extracellular vesicles containing Fe-loaded ferritin (doi: 10.15430/JCP.2021.26.4.244). The authors themselves postulated for the first time an alteration of lung iron homeostasis after asbestos exposure relying on elevated ferritin levels in bronchoalveolar lavage of exposed individuals (doi: 10.1089/ars.2007.1909).

Furthermore, recently Zangari et al (https://doi.org/10.1080/15287394.2022.2164391), have shown that asbestos fiber exert themselves a iron (II) oxidative activity. Since the iron loading process in the ferritin cage is complex (https://doi.org/10.15430/JCP.2021.26.4.244 ) and specific proteins are required for transporting iron (II) to ferritin cage for oxidation and loading. Therefore the iron oxidative activity of the fibers may compete with ferritin loading, favors free iron binding on the fiber surface and inhibit ferritin derived iron availability (increasing iron deficiency) in a context of iron overload.

In the reviewer opinion the sequestration of ferritin by fibers may represent even a further element contributing to the change of iron homeostasis, and cannot be forgotten. Even if they are quoted, the findings of other authors that the synthesis of ferritin is increased in different cell types exposed to asbestos should also been deeply discussed.

Response to reviewer’s comments: As noted by the reviewer, the authors communicated a similar opinion several decades ago with ferritin being involved (Ghio AJ, Churg A, Roggli VL. Ferruginous bodies: implications in the mechanism of fiber and particle toxicity. Toxicol Pathol 2004; 32(6):643-9. doi: 10.1080/01926230490885733). However, the more recent data does not support an involvement of ferritin in the formation of asbestos bodies. This investigation was previously included in this submitted manuscript (THE FORMATION OF FERRUGINOUS BODIES AND IRON):

“Proteins and mucopolysaccharides have also been proposed as possible components of an asbestos body (99). Comparable to metal cations, quarternary ammonium groups of a protein can be adsorbed onto negatively charged portions of a surface. Subsequently, the adsorption of positively charged proteins would localize to those segments of the fiber surface with greater silanol concentrations as a result of the negative charge. The inclusion of protein is anticipated to be indiscriminate and of little consequence to the organization of the asbestos body. In contrast to this, it has been proposed that the coat includes ferritin since the inorganic core of this storage protein can contain crystalline iron particles in the same size range (99, 116). Analytical comparison supported ferritin (or possibly hemosiderin) in an aggregated and/or misfolded form being a component of the asbestos body (99). While expression is increased in cells exposed to asbestos, ferritin does not stain with Perls’ Prussian blue (9, 126-129). In addition, immunohistochemistry does not confirm the presence of ferritin in the coat of asbestos bodies (45). The formation of the crystalline material in an asbestos body has similarly been attributed to hemosiderin (19). Hemosiderin is an iron storage protein which results after incomplete degradation of ferritin, has a higher iron-to-protein ratio, is less soluble in aqueous solutions, and is considered a more stable and less available form of metal. However, the process of asbestos body formation occurs in the lysosome which is normally the site of degradation of both ferritin and hemosiderin to iron (130). It is problematic to characterize either ferritin or hemosiderin in the surface coat with the formation of the asbestos body occurring at the cellular site of ferritinophagy.”

Furthermore, ferruginous and asbestos bodies are products of iron biomineralization. There many examples of iron biomineralization. Two of these were previously included in the submitted manuscript (THE FORMATION OF FERRUGINOUS BODIES AND IRON):

“Finally, in a pathway of iron biomineralization approximating asbestos body formation, several bacteria incorporate iron from their environment to synthesize intracellular nanoparticles of magnetite or greigite (Fe3S4) in organelles called magnetosomes (138, 139). This biomineralization also occurs within intracellular membranous vesicles that originate from invaginations of the cytoplasmic membrane (140). The bacteria take up either dissolved ferrous or ferric cation from the environment, store it in the cell as Fe3+, reduce it for trafficking to magnetosomes, and precipitate it as magnetite after oxidation of Fe2+. Precursors include oxidized phases including ferrihydrite. In another related pathway of iron biomineralization, iron plaque formation in plants is dependent on the availability of Fe2+ which reacts with oxygen to generate Fe3+ oxyhydroxides and deposit on root surfaces (141, 142). Such iron plaque includes crystalline iron oxyhydroxides and demonstrates positive staining with Perls’ Prussian blue (143, 144). On the plant roots, iron plaque can function to bind other metals (e.g., cadmium, lead, zinc, and aluminum) impeding their entry (143). These iron bio-mineralization pathways (magnetosomes and iron plaque) share features with asbestos bodies including the reduction and oxidation of iron, precipitation of iron oxides, and adsorption of other metals.”

These examples of biomineralization all share similarities with some substance providing an interface which binds iron, an oxidation of ferrous cation, and precipitation of ferrihydrite and other iron oxides later (Chan CS, Fakra SC, Edwards DC, Emerson D, Banfield JF. Iron oxyhydroxide mineralization on microbial extracellular polysaccharides. Geochimica et Cosmochimica Acta 2009; 73: 3807-3818). However, none on these other types of iron biomineralization focus on a significant participation by ferritin. This data on the description of biomineralization in microbes and plants does not support a central involvement of ferritin in other forms of biomineralization including ferruginous and asbestos bodies in humans.

Reviewer’s comment: (ii) The second point regards the role played by FB in the asbestos-related diseases. Is it a defense mechanism? Or can it contribute to the cell/tissue damage? Some authors have provided evidence that asbestos bodies really can propagate damage and stimulate the inflammatory process, even if, from some points of view can inhibit the cytotoxic power of these peculiar structures. I think that the latter possibility deserves to be discussed. Some researchers  presented evidence that FB can contribute to maintain the toxicity of AB and their stimulus to chronic inflammation. They can damage DNA (https://doi.org/10.1080/15287394.2023.2181899; https://doi.org/10.1080/15287394.2012.688478). Even if these findings are in some cases debatable (DNA damage) and waiting to be confirmed in “in vivo” models, they must be mentioned and discussed in a complete review.

Response to reviewer’s comments: Whether individual asbestos bodies or a group/collection of asbestos bodies will show increased effect relative to the fiber itself will depend on:

  1. The number of functional groups on the asbestos surface which are not actively complexing iron.
  2. The number of functional groups of the asbestos surface which are not actively complexing an alternative metal cation
  3. The endpoint measured.

The focus of this review is the relationship between asbestos and iron. Some portion of the manuscript includes the biological effect of asbestos and its relationship with iron availability and the formation of ferruginous bodies and iron. The manuscript describes both 1) the biological effect of asbestos and its relationship with iron availability and 2) formation of an asbestos body to reflect a capacity of the surface to complex iron and provide a template for production of ferrihydrite and other iron oxides; the latter is reduced to chemistry/biochemistry. It is unclear whether this is a defense mechanism or a contributor to cell/tissue damage. Subsequently, the manuscript does not conclude on either.

Reviewer’s comment: Page 9, Figure 5. Please indicate which Fe exporter would be blocked. Please see also the previous comment on hephaestin/ferroportin-mediated iron export.

Response to reviewer’s comments: The only exporter recognized is ferroportin-1 although possible metal release through extracellular vesicles is currently being delineated.

The manuscript has been amended to read (FIGURE 5 LEGEND):

“Asbestos body development and iron. In the phagolysosome, asbestos complexes cell iron impacting a functional iron deficiency (A). The host cell responds with increases in the import (e.g. that by divalent metal transporter 1) and decreases in the export (i.e. that by ferroportin 1) of iron elevating storage of the metal in ferritin (designated by red dots).”

Reviewer’s comment: In the reviewer opinion the quality of the images is quite poor and they add nothing to the text, it is recommended to improve them or remove them altogether.

Response to reviewer’s comments: Image quality was lost with pasting images in the provided template. Images are now attached as JPEGs for the reviewer to better judge quality.

Reviewer’s comment: Page 10, line 389. Please note that recently Bardelli and colleagues (DOI: 10.1007/s10653-023-01557-0) reported in the FB coating “the presence of ferrihydrite, and, to a lower extent, of goethite, as the major phases, and the absence of hematite”. This reference should be added.

Response to reviewer’s comments: The manuscript has been amended to included (THE FORMATION OF FERRUGINOUS BODIES AND IRON):

“Analysis supports that the coat on the asbestos bodies has a high iron concentration, that most of the metal is oxidized (i.e. ferric), and the composition is similar to ferrihydrite (99, 106-108,

Bardelli F, Giacobbe C, Ballirano P, Borelli V, Di Benedetto F, Montegrossi G, Bellis D, Pacella A. Closing the knowledge gap on the composition of the asbestos bodies. Environ Geochem Health 2023; 45(7):5039-5051. doi: 10.1007/s10653-023-01557-0).

Reviewer’s comment: Page 12, References

Reference 101: in the present review, the author mention that : iron in the coat of AB does not appear to directly participate in the generation of reactive oxygen species (ref 101).  In that paper Bardelli et al. don’t  assert this. Rather they say: “indicated that hematite and metallic iron, whose presence in the AB was claimed in a previous study, are absent or present in amounts well below 5%, and thus that the AB are mainly composed by ferritin and/or hemosiderin”.

Response to reviewer’s comments: In the submitted manuscript, the last citation of seven citations to Bardelli et al. was incorrectly employed to support the statement “Iron in the coat of asbestos bodies does not appear to directly participate in the generation of reactive oxygen species”. The manuscript has been amended (THE FORMATION OF FERRUGINOUS BODIES AND IRON):

“It has been suggested that the coat in the asbestos body is a protective mechanism. There is a sequestration of the iron as oxyhydroxides in which the valence sites of the metal are fully coordinated and unavailable for electron exchange (e.g., free radical generation). There is a diminished generation of oxidative stress and a reduced toxicity of coated fibers relative to uncoated fibers (20,133). Iron in the coat of asbestos bodies, that is iron oxy oxyhydroxides, may not directly participate in the generation of reactive oxygen species. Exposures of animals and humans to equivalent iron oxides (intranasal, intratracheal, intravenous, and inhaled) are most frequently without biological effects (134-137). In medicine, iron oxide nanoparticles are widely used as therapeutic, delivery and diagnostic agents.”

Manuscript ID: ijms-2514248 – Reviewer 1

Reviewer’s comment: This review from Ghio and colleagues focuses on the participation of iron in the pathogenesis of asbestos related diseases. In their introduction the authors, who represent excellence in iron related asbestos toxicology research, state that “theories of disease pathogenesis  following asbestos exposure were developed which focused on a participation of iron” … one suggesting “that pathogenesis of asbestos-related disease is dependent on the complexation chemistry of iron” and the other on its “oxidant chemistry” mainly through “an in vivo production of  hydroxyl radical by iron”. Anyway in this review they pursue only the first theory, in the light of which they start from a detailed description of iron chemistry and reactivity of asbestos fibers, subsequently the consequences of asbestos internalization by macrophages (only) on cellular iron homeostasis are reported and finally the formation of asbestos bodies is discussed.

In the reviewer opinion the two theories are not mutually exclusive and should both be discussed. Alternatively the title of the review should be reconsidered, as for example: "Role of asbestos-induced cellular iron deficiency in the pathogenesis of asbestos related diseases".

Furthermore the perspective supported by Ghio and colleagues in the present review is quite different from that in 2004 (Ghio et al. DOI: 10.1080/01926230490885733). Quoted verbatim: “IMPLICATIONS Free radical generation by the fiber and particle is mediated in some part by coordinated metal. Ferruginous bodies represent an attempt by the host to sequester the metal adsorbed to the surface of a fiber and diminish the oxidative challenge presented by a fiber or particle. While much of the iron coordinated onto the surface will be stored in a catalytically less reactive state within ferritin included in the ferruginous body, some portion of this metal will ultimately be catalytically active and therefore capable of supporting the generation of an oxidative stress. Consequently, the observation of ferruginous bodies on microscopic inspection of lung tissue should be interpreted as supporting a specific mechanism of injury (i.e., a metal-catalyzed oxidative stress). This would apply not only to tissue injury after exposure to fibers and particles but also is pertinent to damage associated with surfaces of many appliances placed in the body if that surface presents oxygen-containing functional groups (Jordan et al., 2002)”.

This change of view may have a scientific basis, which the reviewer thinks should be justified.

Response to reviewer’s comments: The reviewer recommends that the current manuscript also include focus to the “oxidant chemistry” of iron and this can involve “an in vivo production of  hydroxyl radical by iron”. Several decades of science (two since the perspective quoted by the reviewer: Ghio AJ, Churg A, Roggli VL. Ferruginous bodies: implications in the mechanism of fiber and particle toxicity. Toxicol Pathol 2004; 32(6):643-9. doi: 10.1080/01926230490885733) support an involvement of complexation chemistry of iron in the biological effects of asbestos. This involvement of the complexation chemistry of iron in the biological effect of asbestos is addressed throughout the entirety of this submitted manuscript. However, regarding inclusion of the oxidative chemistry of iron in the biological effects of particles and fibers (e.g. asbestos), the authors have more recently stated that:

“Life is ferrocentric with iron being an essential micronutrient required by every cell. A favorable oxidation-reduction potential and a relative abundance led to its evolutionary selection for a wide range of fundamental functions. Following the introduction of oxygen to the atmosphere as a product of photosynthesis, water-soluble ferrous ion (Fe2+) was effectively removed from the Earth’s crust. The resultant ferric ion (Fe3+) remained but, being insoluble in water, at concentrations inadequate to meet the requirements for life; the concentrations of Fe3+ in water at physiologic pH values approximate 10−18 M while those required for life approach 10−6 M. Accordingly, greater quantities of metal had to be procured to support emerging life. This challenge was realized by living systems by utilizing two major pathways to acquire essential iron: 1) the chemical reduction of Fe3+ to Fe2+ (i.e. ferrireduction) with its subsequent import and utilization and 2) the complexation of Fe3+ with chelators coupled with receptors for uptake of the complex and employment of the metal. In addition to solubility limiting its availability, iron-catalyzed generation of radicals presented a potential for oxidative stress; improperly sequestered iron has a theoretical potential to catalyze toxic reactive oxygen species. Such reactivity mandated that iron homeostasis be tightly controlled. Living systems evolved strategies to regulate the procurement of adequate iron for cellular function and homeostasis precluding damage to biological macromolecules. The import, storage, and efflux of this metal is vigilantly regulated. Accordingly, life exists at the interface between iron-deficiency and iron-sufficiency.” (Ghio AJ, Soukup JM, Dailey LA, Madden MC. Air pollutants disrupt iron homeostasis to impact oxidant generation, biological effects, and tissue injury. Free Radic Biol Med. 2020;151:38-55. doi: 10.1016/j.freeradbiomed.2020.02.007.)

Life is positioned at an interface between iron deficiency and sufficiency. This is evident with a significant proportion of the human population defined to be iron deficient (e.g., toddlers and children of preschool age internationally have a rate of iron deficiency anemia that can approximate 50%). Iron overload (focal or otherwise) and a possible support of an uncontrolled production of hydroxyl radical is extremely rare in any living system, including humans, if it does occur. The control of the potential for iron to support such oxidant generation can reflect design. The intracellular concentration of iron can approximate 1 to 10 µM (this is total). An empty or labile coordination site on the iron cation is required to support electron exchange (e.g., hydroxyl radical generation). The intracellular concentration of compounds and substances that complex and assume empty or labile coordination sites on iron (e.g. phosphates and carboxylates) can be millions of times higher than this. Under these conditions, an empty or labile coordination site is precluded and conditions which might support generation for hydroxyl radical do not exist (again, this is in a living system). Subsequently, with this background and the data presented in the submitted manuscript, the authors cannot fairly address the “oxidant chemistry” of iron in the biological effect of asbestos. Much of this data has been made available in the last few decades.

The association between asbestos and iron is comparable to disease in many fields of medicine (Beavers CJ, Ambrosy AP, Butler J, Davidson BT, Gale SE, Piña IL, Mastoris I, Reza N, Mentz RJ, Lewis GD. Iron Deficiency in Heart Failure: A Scientific Statement from the Heart Failure Society of America. J Card Fail. 2023; 29(7):1059-1077. doi: 10.1016/j.cardfail.2023.03.025.)

Reviewer’s comment: The reviewer also found some shortcomings and wrong attribution (later discussed in the specific comments) regarding the references of some statements. In general, the reviewer found this review too focused on the theory currently supported by the authors, so as not to maintain the expectations generated by the title.

In this form and without discussing the issues raised by the reviewer this review is not suitable for publication, unless the subject dealt with is limited to the discussion of the authors theory about the role played by iron sequestration in the induction of asbestos-related diseases.

Response to reviewer’s comments: Based on these specific comments provided by the reviewer, the manuscript has been amended to include (TITLE):

“Asbestos and the complexation chemistry of iron”

Reviewer’s comment: Specific comments. Page 1, line 57. Iron involvement in the biological effects of asbestos is also supported by some genetic studies investigating the role of iron related gene polymorphisms in the etiology of asbestos related cancers (mesothelioma, MPM and lung cancer, LC). These studies report three SNPs, localized in the ferritin heavy polypeptide, transferrin, and hephaestin genes,  significantly associated with protection against development of MPM and LC (DOI: 10.1080/15287394.2015.1123452 and  doi: 10.1080/15287394.2019.1694612). Iron signature in asbestos-related disease should be also discussed.

Response to reviewer’s comments: The manuscript has been amended to include (INTRODUCTION):

“There is support for metal involvement in the biological effect of asbestos with inclusion of iron-related proteins in the genetic signatures of asbestos-related malignancies and both chelation therapy and phlebotomy demonstrating possible preventive effects against asbestos-induced carcinogenesis in animal models (5-9).”

The following references have been added (REFERENCES):

Jiang L, Akatsuka S, Nagai H, Chew S-H, Ohara H, Okazaki Y et al. Iron overload signature in chrysotile-induced malignant mesothelioma. J Pathol 2012 Nov;228(3):366-77. doi: 10.1002/path.4075.

Crovella S, Bianco AM, Vuch J, Zupin L, Moura RR, Trevisan E et al. Iron signature in asbestos-induced malignant pleural mesothelioma: A population-based autopsy study. J Toxicol Environ Health A 2016;79(3):129-41. doi: 10.1080/15287394.2015.1123452.

Celsi F, Crovella S, Moura RR, Schneider M, Vita F, Finotto L, et al. Pleural mesothelioma and lung cancer: the role of asbestos exposure and genetic variants in selected iron metabolism and inflammation genes. J Toxicol Environ Health A. 2019;82(20):1088-1102. doi: 10.1080/15287394.2019.1694612.

Reviewer’s comment: Page 5, figure 2. The process by which the iron would enter the phago-lysosome (through which carrier?) should be better explained.

Response to reviewer’s comments: With exposure to asbestos, the fiber is phagocytosed by macrophages. Iron is complexed by the fiber. This must occur intracellularly with the asbestos in the phago-lysosome since 1) asbestos in macrophages is localized to phago-lysosomes and 2) there is no extracellular iron available to be complexed by the fiber.

However, the manuscript does not specify how the metal is complexed to the asbestos surface in the phago-lysosome. The required investigation has not been accomplished to define this. With normal import of iron utilizing transferrin-Fe3+, there is acidification of the lysosome, release of the Fe3+, reduction of the Fe3+ to Fe2+ by STEAP, and movement of the Fe2+ across the membrane by DMT1. It is also recognized that NRAMP1 can facilitate transport of iron in the macrophage (Soe-Lin S, Apte SS, Mikhael MR, Kayembe LK, Nie G, Ponka P. Both Nramp1 and DMT1 are necessary for efficient macrophage iron recycling. Exp Hematol. 2010; 38(8):609-17. doi: 10.1016/j.exphem.2010.04.003). However, both DMT1 and NRAMP1 can transport the metal out of the phago-lysosome.

It is possible that functional groups at the surface of the fiber complex available metal which is in the immediate environs. A cascade effect would follow with those molecules/substances/organelles which lost metal then complexing adjacent sources of metal. Accordingly, metal accumulates on the asbestos surface and in the phago-lysosome and this would be associated with the observed deficiency in the cell and mitochondrial dysfunction (reflecting the loss of iron).

Reviewer’s comment: Page 6, line 182. The authors describe only the increase in iron import, but the cells could also limit the export of iron through the ferroportin/Hephaestin (or ceruloplasmin) system, as occurs (thanks to hepcidin activity) during inflammation. A variant of Hephaestin has been associated with a lower risk of developing related asbestos neoplasms. The authors should discuss this point (iron export) also at Page 7 line 256 and figure 5.

Response to reviewer’s comments: There is a scarcity of investigation regarding cellular export of iron. Until recently, the single iron exporter recognized has been ferroportin-1 (FPN1) whose activity is coupled with hephaestin or ceruloplasmin oxidation. Only in the last few years has extracellular vesicles with inclusion of ferritin been considered as an additional potential pathway of iron export.

Regarding ferroportin and asbestos, there have been only three articles that we can find. Investigation which has included hephaestin and asbestos is available in only one article (mentioned previously in this response and now referenced in the revised manuscript). It is difficult to provide much discussion on potential iron export after asbestos exposure based on this limited investigation.

The manuscript has been amended to include (“ASBESTOS, CELL IRON HOMEOSTASIS, AND BIOLOGICAL EFFECT”):

“Changes in the expression and activity of the metal exporter ferroportin-1 (FPN1) as well as exported iron concentrations have yet to be fully delineated but are also possible (Ghio A, Tan RJ, Ghio K, Fattman CL, Oury TD. Iron accumulation and expression of iron-related proteins following murine exposure to crocidolite. J Environ Pathol Toxicol Oncol. 2009;28(2):153-62 and Ghio AJ, Pavlisko EN, Roggli VL. Iron and Iron-Related Proteins in Asbestosis. J Environ Pathol Toxicol Oncol 2015;34(4):277-85.)”

Reviewer’s comment: Page 6 line 200. “The electron transport chain in the mitochondria can be a major source of superoxide production (46, 47)” perhaps this statement is true for most cells, anyway in phagocytes the major source of superoxide is the oxidative-burst.

Response to reviewer’s comments: The manuscript has been amended to read (ASBESTOS, CELL IRON HOMEOSTASIS, AND BIOLOGICAL EFFECT): “The host also attempts to re-acquire its metal directly sequestered by the fiber. Movement of iron necessitates ferrireduction to chemically reduce the metal to the ferrous state (e.g., transmembrane transport). This chemical reduction is frequently accomplished by superoxide generated in the cell (e.g., membranes of the cell, mitochondria, and lysosomes). The major source of superoxide after particle exposure is the phagocyte-associated NADPH oxidoreductase (Ghio AJ, Soukup JM, Stonehuerner J, Tong H, Richards J, Gilmour MI, Madden MC, Shen Z, Kantrow SP. Quartz Disrupts Iron Homeostasis in Alveolar Macrophages To Impact a Pro-Inflammatory Effect. Chem Res Toxicol 2019; 32(9):1737-1747. doi: 10.1021/acs.chemrestox.8b00301). An increased availability of iron (i.e. cell pretreatment with FAC) diminishes a functional iron deficiency and superoxide production following asbestos exposure (41). The “oxidative stress” after asbestos exposure includes mitochondria as a source since rotenone inhibits some portion of superoxide generation and iron uptake by the cell. The electron transport chain in the mitochondria can be an additional source of superoxide production (46, 47). Through a comparable pathway, cellular superoxide generation can increase after exposure to iron chelators (48, 49). The generated superoxide functions in ferrireduction with cell acquisition, transport, and translocation of iron. Such ferrireduction is an essential, and frequently limiting, reaction in iron acquisition, transport, and translocation (50, 51).”

Reviewer’s comment: In this context the reviewer believes that the role of neutrophils should also be considered, as they engulf asbestos fibers (personal communication), produce more superoxide than macrophages, release, upon activation,  lactoferrin and elastase, which the authors argue may release the iron bound to elastase (page 7, lines 246-248). In figure 4 these cells are represented within the alveolus, but their role is not dissected.

Response to reviewer’s comments: To fairly address the integration of the roles of neutrophils and macrophages in impacting iron homeostasis after exposures to particles (including fibrous particles such as asbestos) and infectious agents necessitates its own article. Numerous products of neutrophils and macrophages are focused on a mobilization of the metal (e.g., proteases), reduction of the metal (e.g., superoxide and its related products), and translocation back to the host (e.g., lactoferrin). A complete discussion of these multiple pathways is not possible here.

The manuscript has been amended to include (FIGURE 4 LEGEND): “The biological effect of asbestos and iron. Decreased iron availability affects kinase and transcription factor activation which coordinate a release of mediators relevant to inflammatory and fibrotic responses. An influx of inflammatory cells (e.g., macrophages and neutrophils) corresponds with the decreased metal availability. Products from the phagocytic cells can impact iron homeostasis with mobilization (e.g., proteases), ferrireduction (e.g., superoxide), and translocation (e.g., lactoferrin) of the metal allowing the host to restrict its availability. There is also an increased number of fibroblasts and a deposition of collagen (designated by blue helical units), elastin (designated by yellow units), and extracellular polymeric substances which correspond to the functional iron deficiency.”

Reviewer’s comment: Furthermore the effects of asbestos internalization on iron homeostasis of mesothelial cells (which display phagocytic function DOI: 10.1097/IGC.0000000000000697 and differentiate into macrophage-like cells DOI: 10.1111/j.1600-0463.2011.02803.x) should be also discussed since the pathogenesis of chrysotile-induced mesothelial carcinogenesis seems closely associated with local iron overload (doi: 10.1002/path.4075; DOI: 10.1038/bjc.1989.344).

Response to reviewer’s comments: The manuscript has been included to include this (ASBESTOS, CELL IRON HOMEOSTASIS, AND BIOLOGICAL EFFECT).

“Pleural inflammation and fibrosis (e.g., pleural effusions and plaques respectively) will follow lymphatic translocation of the asbestos to the membranes enveloping the lungs through identical or similar mechanistic pathways. However, in addition to macrophages and neutrophils, it is possible that mesothelial cells can contribute to the biological effect after differentiation to phagocytic cells and anti-gen presenting cells.”

The following references have been added (REFERENCES):

“Katz S, Balogh P, Kiss AL. Mesothelial cells can detach from the mesentery and differentiate into macrophage-like cells. APMIS  2011; 119(11):782-93. doi: 10.1111/j.1600-0463.2011.02803.x

Shaw TJ, Zhang XY, Huo Z, Robertson D, Lovell PA, Dalgleish AG, Barton DP. Human Peritoneal Mesothelial Cells Display Phagocytic and Antigen-Presenting Functions to Contribute to Intraperitoneal Immunity. Int J Gynecol Cancer 2016 Jun;26(5):833-8. doi: 10.1097/IGC.0000000000000697”

The last two references provided by the reviewer are not added to the manuscript as they are considered by the authors to misinterpret the exposure to an iron chelator as an “overload”.

Reviewer’s comment: Page 7 line 215-232. “Inflammation is included in the host response to a functional iron deficiency”. This statement is debatable and should be reconsidered, since inflammation itself (via hepcidin), induces systemic iron deficiency and ferritin is an acute phase protein. The role of iron deficiency in the induction of inflammation is mostly supported by indirect studies, when phagocytosis of the fibers by the phagocytes itself induces the release of inflammatory mediators.

Response to reviewer’s comments: The article carefully expresses a relationship of inflammation with asbestos and iron availability:

“Inflammation is included in the host response to a functional iron deficiency. The complexation of host iron by asbestos initiates pathways which culminate in a release of inflammatory mediators and inflammation. Cell iron deficiency following exposures to fibers activates kinases and transcription factors which are associated with a release of inflammatory mediators. Cell exposure to asbestos activates mitogen-activated protein (MAP) kinases and some portion of this response can be diminished by increasing the concentration of available iron (52-54). Transcription factors involved in expression of inflammatory mediators are also activated by particle/fiber exposure (55-58). Many of these same transcription factors control cell death and if the response to correct the functional iron deficiency is insufficient, some form of apoptosis will ensue (59-62). Comparable to MAP kinases, increased iron availability decreases the activation of transcription factors following exposures to particle/fiber (53, 54, 63). After impacting cell signaling and transcription factors, asbestos exposure will produce changes in expression of inflammatory mediators. Changes in protein expression for IL-6 and IL-8 after asbestos exposure are diminished by cell pretreatment with iron reflecting their association with metal availability (64). It is the functional iron deficiency after asbestos exposure which activates kinase signaling and transcription, increases release of mediators, and coordinates the inflammatory response.”

It is unclear what the reviewer is referring to as “debatable”. The references support the relationship between the cell and molecular basis for inflammation and iron availability. Changes in mediators (e.g., hepcidin), decreased serum iron levels, and anemias of chronic disease are products of the cell and molecular events. This is comparable to numerous other compounds and substances which complex iron to decrease availability of the metal and, like asbestos, are associated with inflammation, fibrosis, and neoplasm (e.g. other particles and fibers, microbials, bleomycin and doxorubicin). More references can be provided specifically on the relationship between inflammation and iron availability if greater support is thought necessary.

Reviewer’s comment: Cellular iron deficiency may help to explain the inflammation which generate a pro-inflammatory microenvironment favoring cancer genesis, anyway radical generation (presumably via catalytic action by the Fenton reaction induced by iron complexation by the fiber surface) is necessary for neoplastic transformation to occur, and authors should also discuss this issue.

Response to reviewer’s comments: This theory of inflammation and carcinogenesis is unsupported by several decades of investigation. The involvement of hydroxyl radical in the biological effects of asbestos is difficult to postulate. This requires empty or labile coordination sites to support electron transport. Iron in a living system is tightly complexed and carefully handled/transported. In a living system, the ratio of compounds and substances which complex iron to the metal absolutely precludes a generation of hydroxyl radical via the Fenton reaction. In any cell, the ratio of phosphates and carboxylates alone is 106 greater than iron.

The manuscript has been amended to include a discussion of the relationships between asbestos, iron, and carcinogenesis (ASBESTOS, CELL IRON HOMEOSTASIS, AND BIOLOGICAL EFFECT):

“Finally, malignancy is a biological effect of asbestos. A diverse group of cells in a tissue, each with its individual requirements for iron and pathways for metal acquisition, has access to a pool of iron. Comparable to exposures to numerous compounds and substances associated with carcinogenesis (e.g. cigarette smoke particle, erionite, and metals), asbestos is associated with a disruption in homeostasis and a functional deficiency of iron. In an environment in which iron is a limited resource, competition will arise between the cells for the metal. The functional iron deficiency after exposure to asbestos can apply a selective pressure on cell communities. Pathways for iron acquisition employed by a malignant cell allow it to compete more successfully for this nutrient which is essential for proliferation and growth (e.g., increased expression of iron importers such as transferrin and lactate and their respective receptors). A malignant cell subsequently will surpass a normal cell in the capacity to proliferate and grow in the iron-depleted environment. By competing more effectively for requisite iron, malignant cells will displace others and dominate the cell population.”

Reviewer’s comment: Page 8,line 174: The formation of ferruginous bodies (FB)

General comment: the authors  disregard two key points in the biology of FB.

(i)                 First the by now acquired certainty that asbestos fibers can absorb many different proteins for example in: https://doi.org/10.1111/j.1349-7006.2011.02087.x; https://doi.org/10.3390/ijerph15010104). Ferritin, one of the protein absorbed by asbestos (10.1080/00984109708984069) is presently recognized as a major component of FB (https://doi.org/10.1080/15287390701380906; DOI: 10.1038/srep44862) together with a minor amount of free iron. So, showing the presence of ferritin in FB is not problematic (page 11, line 434), it is a fact.

Ferritin plays a key role in FB formation, where  this protein is absorbed by the fibers (10.1080/15287394.2022.2164391),  is continuously synthetized and  is also secreted in extracellular vesicles. Indeed asbestos exposed alveolar macrophages are characterized by a high turnover due to the high level of ferroptotic cell death. Upon ferroptosis macrophages release extracellular vesicles containing Fe-loaded ferritin (doi: 10.15430/JCP.2021.26.4.244). The authors themselves postulated for the first time an alteration of lung iron homeostasis after asbestos exposure relying on elevated ferritin levels in bronchoalveolar lavage of exposed individuals (doi: 10.1089/ars.2007.1909).

Furthermore, recently Zangari et al (https://doi.org/10.1080/15287394.2022.2164391), have shown that asbestos fiber exert themselves a iron (II) oxidative activity. Since the iron loading process in the ferritin cage is complex (https://doi.org/10.15430/JCP.2021.26.4.244 ) and specific proteins are required for transporting iron (II) to ferritin cage for oxidation and loading. Therefore the iron oxidative activity of the fibers may compete with ferritin loading, favors free iron binding on the fiber surface and inhibit ferritin derived iron availability (increasing iron deficiency) in a context of iron overload.

In the reviewer opinion the sequestration of ferritin by fibers may represent even a further element contributing to the change of iron homeostasis, and cannot be forgotten. Even if they are quoted, the findings of other authors that the synthesis of ferritin is increased in different cell types exposed to asbestos should also been deeply discussed.

Response to reviewer’s comments: As noted by the reviewer, the authors communicated a similar opinion several decades ago with ferritin being involved (Ghio AJ, Churg A, Roggli VL. Ferruginous bodies: implications in the mechanism of fiber and particle toxicity. Toxicol Pathol 2004; 32(6):643-9. doi: 10.1080/01926230490885733). However, the more recent data does not support an involvement of ferritin in the formation of asbestos bodies. This investigation was previously included in this submitted manuscript (THE FORMATION OF FERRUGINOUS BODIES AND IRON):

“Proteins and mucopolysaccharides have also been proposed as possible components of an asbestos body (99). Comparable to metal cations, quarternary ammonium groups of a protein can be adsorbed onto negatively charged portions of a surface. Subsequently, the adsorption of positively charged proteins would localize to those segments of the fiber surface with greater silanol concentrations as a result of the negative charge. The inclusion of protein is anticipated to be indiscriminate and of little consequence to the organization of the asbestos body. In contrast to this, it has been proposed that the coat includes ferritin since the inorganic core of this storage protein can contain crystalline iron particles in the same size range (99, 116). Analytical comparison supported ferritin (or possibly hemosiderin) in an aggregated and/or misfolded form being a component of the asbestos body (99). While expression is increased in cells exposed to asbestos, ferritin does not stain with Perls’ Prussian blue (9, 126-129). In addition, immunohistochemistry does not confirm the presence of ferritin in the coat of asbestos bodies (45). The formation of the crystalline material in an asbestos body has similarly been attributed to hemosiderin (19). Hemosiderin is an iron storage protein which results after incomplete degradation of ferritin, has a higher iron-to-protein ratio, is less soluble in aqueous solutions, and is considered a more stable and less available form of metal. However, the process of asbestos body formation occurs in the lysosome which is normally the site of degradation of both ferritin and hemosiderin to iron (130). It is problematic to characterize either ferritin or hemosiderin in the surface coat with the formation of the asbestos body occurring at the cellular site of ferritinophagy.”

Furthermore, ferruginous and asbestos bodies are products of iron biomineralization. There many examples of iron biomineralization. Two of these were previously included in the submitted manuscript (THE FORMATION OF FERRUGINOUS BODIES AND IRON):

“Finally, in a pathway of iron biomineralization approximating asbestos body formation, several bacteria incorporate iron from their environment to synthesize intracellular nanoparticles of magnetite or greigite (Fe3S4) in organelles called magnetosomes (138, 139). This biomineralization also occurs within intracellular membranous vesicles that originate from invaginations of the cytoplasmic membrane (140). The bacteria take up either dissolved ferrous or ferric cation from the environment, store it in the cell as Fe3+, reduce it for trafficking to magnetosomes, and precipitate it as magnetite after oxidation of Fe2+. Precursors include oxidized phases including ferrihydrite. In another related pathway of iron biomineralization, iron plaque formation in plants is dependent on the availability of Fe2+ which reacts with oxygen to generate Fe3+ oxyhydroxides and deposit on root surfaces (141, 142). Such iron plaque includes crystalline iron oxyhydroxides and demonstrates positive staining with Perls’ Prussian blue (143, 144). On the plant roots, iron plaque can function to bind other metals (e.g., cadmium, lead, zinc, and aluminum) impeding their entry (143). These iron bio-mineralization pathways (magnetosomes and iron plaque) share features with asbestos bodies including the reduction and oxidation of iron, precipitation of iron oxides, and adsorption of other metals.”

These examples of biomineralization all share similarities with some substance providing an interface which binds iron, an oxidation of ferrous cation, and precipitation of ferrihydrite and other iron oxides later (Chan CS, Fakra SC, Edwards DC, Emerson D, Banfield JF. Iron oxyhydroxide mineralization on microbial extracellular polysaccharides. Geochimica et Cosmochimica Acta 2009; 73: 3807-3818). However, none on these other types of iron biomineralization focus on a significant participation by ferritin. This data on the description of biomineralization in microbes and plants does not support a central involvement of ferritin in other forms of biomineralization including ferruginous and asbestos bodies in humans.

Reviewer’s comment: (ii) The second point regards the role played by FB in the asbestos-related diseases. Is it a defense mechanism? Or can it contribute to the cell/tissue damage? Some authors have provided evidence that asbestos bodies really can propagate damage and stimulate the inflammatory process, even if, from some points of view can inhibit the cytotoxic power of these peculiar structures. I think that the latter possibility deserves to be discussed. Some researchers  presented evidence that FB can contribute to maintain the toxicity of AB and their stimulus to chronic inflammation. They can damage DNA (https://doi.org/10.1080/15287394.2023.2181899; https://doi.org/10.1080/15287394.2012.688478). Even if these findings are in some cases debatable (DNA damage) and waiting to be confirmed in “in vivo” models, they must be mentioned and discussed in a complete review.

Response to reviewer’s comments: Whether individual asbestos bodies or a group/collection of asbestos bodies will show increased effect relative to the fiber itself will depend on:

  1. The number of functional groups on the asbestos surface which are not actively complexing iron.
  2. The number of functional groups of the asbestos surface which are not actively complexing an alternative metal cation
  3. The endpoint measured.

The focus of this review is the relationship between asbestos and iron. Some portion of the manuscript includes the biological effect of asbestos and its relationship with iron availability and the formation of ferruginous bodies and iron. The manuscript describes both 1) the biological effect of asbestos and its relationship with iron availability and 2) formation of an asbestos body to reflect a capacity of the surface to complex iron and provide a template for production of ferrihydrite and other iron oxides; the latter is reduced to chemistry/biochemistry. It is unclear whether this is a defense mechanism or a contributor to cell/tissue damage. Subsequently, the manuscript does not conclude on either.

Reviewer’s comment: Page 9, Figure 5. Please indicate which Fe exporter would be blocked. Please see also the previous comment on hephaestin/ferroportin-mediated iron export.

Response to reviewer’s comments: The only exporter recognized is ferroportin-1 although possible metal release through extracellular vesicles is currently being delineated.

The manuscript has been amended to read (FIGURE 5 LEGEND):

“Asbestos body development and iron. In the phagolysosome, asbestos complexes cell iron impacting a functional iron deficiency (A). The host cell responds with increases in the import (e.g. that by divalent metal transporter 1) and decreases in the export (i.e. that by ferroportin 1) of iron elevating storage of the metal in ferritin (designated by red dots).”

Reviewer’s comment: In the reviewer opinion the quality of the images is quite poor and they add nothing to the text, it is recommended to improve them or remove them altogether.

Response to reviewer’s comments: Image quality was lost with pasting images in the provided template. Images are now attached as JPEGs for the reviewer to better judge quality.

Reviewer’s comment: Page 10, line 389. Please note that recently Bardelli and colleagues (DOI: 10.1007/s10653-023-01557-0) reported in the FB coating “the presence of ferrihydrite, and, to a lower extent, of goethite, as the major phases, and the absence of hematite”. This reference should be added.

Response to reviewer’s comments: The manuscript has been amended to included (THE FORMATION OF FERRUGINOUS BODIES AND IRON):

“Analysis supports that the coat on the asbestos bodies has a high iron concentration, that most of the metal is oxidized (i.e. ferric), and the composition is similar to ferrihydrite (99, 106-108,

Bardelli F, Giacobbe C, Ballirano P, Borelli V, Di Benedetto F, Montegrossi G, Bellis D, Pacella A. Closing the knowledge gap on the composition of the asbestos bodies. Environ Geochem Health 2023; 45(7):5039-5051. doi: 10.1007/s10653-023-01557-0).

Reviewer’s comment: Page 12, References

Reference 101: in the present review, the author mention that : iron in the coat of AB does not appear to directly participate in the generation of reactive oxygen species (ref 101).  In that paper Bardelli et al. don’t  assert this. Rather they say: “indicated that hematite and metallic iron, whose presence in the AB was claimed in a previous study, are absent or present in amounts well below 5%, and thus that the AB are mainly composed by ferritin and/or hemosiderin”.

Response to reviewer’s comments: In the submitted manuscript, the last citation of seven citations to Bardelli et al. was incorrectly employed to support the statement “Iron in the coat of asbestos bodies does not appear to directly participate in the generation of reactive oxygen species”. The manuscript has been amended (THE FORMATION OF FERRUGINOUS BODIES AND IRON):

“It has been suggested that the coat in the asbestos body is a protective mechanism. There is a sequestration of the iron as oxyhydroxides in which the valence sites of the metal are fully coordinated and unavailable for electron exchange (e.g., free radical generation). There is a diminished generation of oxidative stress and a reduced toxicity of coated fibers relative to uncoated fibers (20,133). Iron in the coat of asbestos bodies, that is iron oxy oxyhydroxides, may not directly participate in the generation of reactive oxygen species. Exposures of animals and humans to equivalent iron oxides (intranasal, intratracheal, intravenous, and inhaled) are most frequently without biological effects (134-137). In medicine, iron oxide nanoparticles are widely used as therapeutic, delivery and diagnostic agents.”

Reviewer 2 Report

This is a very well written and comprehensive review of asbestos and iron. The concept of functional ion deficiency shown in Figure 3 is an interesting perspective, and I feel that it is well discussed from this point of view as well.

Fibrosis of the lungs, or asbestosis, is also discussed with good logic.

However, I wonder how the presence of iron is related to carcinogenesis from the viewpoint mentioned in this article.

For normal cells that may exist in the vicinity of asbestos fibers and transform into lung cancer, iron on the surface of the fibers itself was thought to be a cancer-causing factor. However, there are many unknowns about mesothelioma.

I would like to see a description of this point as well.

Author Response

Manuscript ID: ijms-2514248 – Reviewer 2

Reviewer’s comment: This is a very well written and comprehensive review of asbestos and iron. The concept of functional iron deficiency shown in Figure 3 is an interesting perspective, and I feel that it is well discussed from this point of view as well.

Fibrosis of the lungs, or asbestosis, is also discussed with good logic.

However, I wonder how the presence of iron is related to carcinogenesis from the viewpoint mentioned in this article.

For normal cells that may exist in the vicinity of asbestos fibers and transform into lung cancer, iron on the surface of the fibers itself was thought to be a cancer-causing factor. However, there are many unknowns about mesothelioma.

I would like to see a description of this point as well.

Response to reviewer’s comments: The manuscript has been amended to include (ASBESTOS, CELL IRON HOMEOSTASIS, AND BIOLOGICAL EFFECT):

“Finally, malignancy is a biological effect of asbestos. A diverse group of cells in a tissue, each with its individual requirements for iron and pathways for metal acquisition, has access to a pool of iron. Comparable to exposures to numerous compounds and substances associated with carcinogenesis (e.g. cigarette smoke particle, erionite, and metals), asbestos is associated with a disruption in homeostasis and a functional deficiency of iron. In an environment in which iron is a limited resource, competition will arise between the cells for the metal. The functional iron deficiency after exposure to asbestos can apply a selective pressure on cell communities. Pathways for iron acquisition employed by a malignant cell allow it to compete more successfully for this nutrient which is essential for proliferation and growth (e.g. increased expression of iron importers such as transferrin and lactate and their respective receptors). A malignant cell subsequently will surpass a normal cell in the capacity to proliferate and grow in the iron-depleted environment. By competing more effectively for requisite iron, malignant cells will displace others and dominate the cell population.”

Round 2

Reviewer 1 Report

The authors partially answered the questions raised by the reviewer, however the reviewer considers these answers sufficient for the publication of what he/she still considers an “Opinion” rather than a “Review”.